# 3D genome alterations associated with dysregulated *HOXA13* expression in high-risk T-lineage acute lymphoblastic leukemia

Lu Yang [1,2,12], Fengling Chen[3,8,9,12], Haichuan Zhu[1,2,10,12], Yang Chen[3,11], Bingjie Dong[1,2], Minglei Shi[3,4], Weitao Wang[1,2], Qian Jiang[5], Leping Zhang[6], Xiaojun Huang[2,5,13 ✉], Michael Q. Zhang[3,4,7,13 ✉] & Hong Wu [1,2,5,13 ✉]

3D genome alternations can dysregulate gene expression by rewiring enhancer-promoter interactions and lead to diseases. We report integrated analyses of 3D genome alterations and differential gene expressions in 18 newly diagnosed T-lineage acute lymphoblastic leukemia (T-ALL) patients and 4 healthy controls. 3D genome organizations at the levels of compartment, topologically associated domains and loop could hierarchically classify different subtypes of T-ALL according to T cell differentiation trajectory, similar to gene expressions-based classification. Thirty-four previously unrecognized translocations and 44 translocation-mediated neo-loops are mapped by Hi-C analysis. We find that neo-loops formed in the non-coding region of the genome could potentially regulate ectopic expressions of *TLX3*, *TAL2* and *HOXA* transcription factors via enhancer hijacking. Importantly, both translocation-mediated neo-loops and *NUP98*-related fusions are associated with *HOXA13* ectopic expressions. Patients with *HOXA11-A13* expressions, but not other genes in the *HOXA* cluster, have immature immunophenotype and poor outcomes. Here, we highlight the potentially important roles of 3D genome alterations in the etiology and prognosis of T-ALL.

[1] The MOE Key Laboratory of Cell Proliferation and Differentiation, School of Life Sciences, Peking University, Beijing, China. [2] Peking-Tsinghua Center for Life Sciences, Beijing, China. [3] MOE Key Laboratory of Bioinformatics, Center for Synthetic and Systems Biology, Bioinformatics Division, BNRist, Department of Automation, Tsinghua University, Beijing, China. [4] School of Medicine, Tsinghua University, Beijing, China. [5] Peking University Institute of Hematology, National Clinical Research Center for Hematologic Disease, Beijing, China. [6] Department of Pediatrics, Peking University People's Hospital, Beijing, China. [7] Department of Biological Sciences, Center for Systems Biology, The University of Texas, Richardson, TX, USA. [8] Present address: Center for Stem Cell Biology and Regenerative Medicine, MOE Key Laboratory of Bioinformatics, Tsinghua University, Beijing, China. [9] Present address: Tsinghua-Peking Center for Life Sciences, Beijing, China. [10] Present address: Institute of Biology and Medicine, College of Life and Health Sciences, Wuhan University of Science and Technology, Hubei, China. [11] Present address: Department of Biochemistry and Molecular Biology, The State Key Laboratory of Medical Molecular Biology, Institute of Basic Medical Sciences, School of Basic Medicine, Chinese Academy of Medical Sciences and Peking Union Medical College, Beijing, China. [12] These authors contributed equally: Lu Yang, Fengling Chen, Haichuan Zhu. [13] These authors jointly supervised this work: Xiaojun Huang, Michael Q. Zhang, Hong Wu. ✉email: xjhrm@medmail.com.cn; michael.zhang@utdallas.edu; hongwu@pku.edu.cn

T-ALL is an aggressive hematological malignancy caused by genetic and epigenetic alterations that affect normal T-cell development[1]. T-ALL represents 15% of pediatric and 25% of adult cases of acute lymphoblastic leukemia (ALL)[1,2], early T-cell precursor ALL (ETP ALL) is a high-risk subtype, which is characterized by an immature immunophenotype and a gene expression profile similar to early T-cell precursors[3,4]. Compared to ETP ALL, non-ETP ALLs, including *HOXA*-, *TLX*-, and *TAL*-related subgroups, are blocked at the later T-cell differentiation stages[5]. Recent whole-exome and RNA sequencing analyses of large T-ALL cohorts have identified driver mutations, dysregulated oncogenic transcription factors, and pathways as the major contributors to its pathogenesis[4,6–8]. However, whether the noncoding region of the genome and 3D-genomic structure play important roles in T-ALL development are largely unknown.

The genomes are hierarchically organized by multi-scaled structural units, including compartments, topologically associated domains (TADs), and loops, which can be identified by Hi-C[9]. At megabase scale, genomes are segregated into A and B compartments, which broadly correspond to transcriptionally active and inactive regions of the genome, respectively[10]. The A and B partitioning of the genome is dynamic, and A-to-B or B-to-A switches have been reported during normal development[11] and in disease states[12]. TADs are genomic regions separated by the binding of insulating proteins, such as CTCF. An important function of TAD is to encompass the enhancers and their controlled gene inside the same domain[9,13], and therefore, DNA elements within the TAD preferentially form intradomain rather than interdomain interactions[13]. At the kilobase scale, linear DNA is folded in loops, probably by loop extrusion, through the action of CTCF and the cohesin complex[14,15]. Loops frequently bridge promoter and enhancer interaction[9] and are further organized into individual TADs[13]. Each of these layers of organization have pronounced effects on gene expression[9,10,16,17]. Recently, Kloetgen et al. discovered that recurrent TAD boundary changes in the *MYC* locus are associated with MYC dysregulation while NOTCH pathway activation can also regulate 3D genome organization in T-ALL[18].

Chromosomal rearrangements are common in cancers and have the potential to disrupt TAD boundaries. Disruption of TAD boundary can create ectopic loops (neo-loop) between enhancers and promoters that are normally separated, termed enhancer hijacking. Enhancer hijacking can result in aberrant gene expression, including ectopic expression of oncogenes[19–26]. However, since enhancer hijacking often happens in the noncoding regions of the genome, it is difficult to identify such events via whole-exome and RNA sequencing analyses.

To determine whether alterations in the 3D genome organization are associated with malignant transformation of T-ALL, we conduct BL-Hi-C[27] analysis using purified primary leukemic blasts from 18 newly diagnosed T-ALL patients, including 8 ETP ALL and 10 non-ETP ALL, two clinical subtypes of T-ALL, as well as normal peripheral T cells from 4 healthy volunteers. The maximum resolutions of the chromatin contact maps for ETP, non-ETP ALL, and normal samples are ~3.5, 3.5, and 10 kb, respectively (Supplementary Data 1). We focus our analysis on chromatin translocations, especially those translocations involving the noncoding regions of the genome. Among the 34 previously unrecognized translocations, we identify recurrent *HOXA13* translocations that cause the "neo-loops" formation. Meanwhile, we discover that T-ALL with *NUP98*-related fusions are associated with enhanced loop structures within the 5′*HOXA* TAD. Taken together, our findings suggest that chromosomal rearrangements can reshape the loop structures of *HOXA* locus in T-ALL by "cis" (enhancer hijacking) and "trans" (oncogenic fusion events) mechanisms. Furthermore, by studying the

association between 3D genome alterations and clinical phenotypes, we find that ectopic expression of the *HOXA11-A13* genes is associated with immature ETP immunophenotype and poor outcome of T-ALL.

## Results

**3D genome landscape in T-ALL**. Principal component analysis (PCA) at the levels of the compartment, TAD, and loop structures demonstrated that the T-ALL samples could be separated from the control samples by PC1, while ETP and non-ETP ALL could be separated by PC2 at all three architectural levels (Fig. 1a, upper panels) and be further delineated by hierarchical clustering analysis (Fig. 1a, lower panels). By detailed comparisons of the 3D chromosomal organizations of the T-ALL samples and the healthy controls, we identified compartment switches corresponding to 3% of genome, ~700 differential TAD boundaries and more than 6000 differential looping events (Supplementary Fig. 1a). These results indicate that there are multi-scaled chromatin structural differences between T-ALLs and normal T cells. Such differences could associate with events leading to T-ALL leukemogenesis or simply reflect the different developmental stages corresponding to T-ALL and normal T cells.

**Correlations between 3D genome alterations and differential gene expressions in T-ALL**. To investigate the potential impact of 3D genome alterations on T-ALL development, we performed RNA-seq analysis on all samples. PCA and hierarchical clustering revealed that the transcriptome changes were highly correlated with that of the 3D genome changes (comparing Fig. 1a, b), similar to recent publication by Kloetgen et al.[18]. By integrating Hi-C and RNA-seq data, we found that a large fraction (996/3392, 29%) of the DEGs was associated with 3D genome alterations (Supplementary Data 2).

Comparing to normal T cells, genes associated with the B-to-A compartment switches, increased domain scores (D-score), and enhanced loops in T-ALL were mostly upregulated (Fig. 1c, red bars and Fig. 1d, "Methods"), and were enriched in pathways such as hematopoietic cell lineage, transcriptional misregulation in cancer, and cell cycle (Supplementary Fig. 1e). In contrast, genes associated with the A-to-B compartment switches, decreased D-scores, and reduced loops in T-ALL were mostly downregulated (Fig. 1c, blue bars and Fig. 1d), and were enriched in pathways such as cytokine–cytokine receptor interaction and T-cell receptor signaling (Supplementary Fig. 1e).

To evaluate the potential impacts of copy-number variations (CNVs) on these 3D genome alteration-associated DEGs, we analyzed CNV data of 242 T-ALL samples from Liu et al.[6] and identified 110 upregulated genes with copy-number gain and 250 downregulated genes with copy-number loss. By aligning these DEGs, we found that only 1/568 3D genome alterations-associated upregulated DEGs exhibited copy-number gain and only 8/428 3D genome alterations-associated downregulated genes had copy-number loss (data not shown). Therefore, majority of the dysregulated genes associated with 3D genome alternation are not due to the CNV changes.

Among the upregulated DEGs, *CDK6* is a potential target for T-ALL treatment[28]. The *CDK6* locus exhibited a strong intra-TAD interaction and its expression was upregulated in all T-ALL samples (Fig. 1e). Similarly, several upregulated oncogenic driver genes and T-ALL-associated transcription factors, such as *MYB, MYCN, BCL11A, SOX4,* and *WT1,* also had increased D-scores (Supplementary Fig. 1f). *SOX4* was a unique case among these dysregulated transcription factors, as Hi-C map showed B-to-A compartment change, increased D-score, and new loop formations between

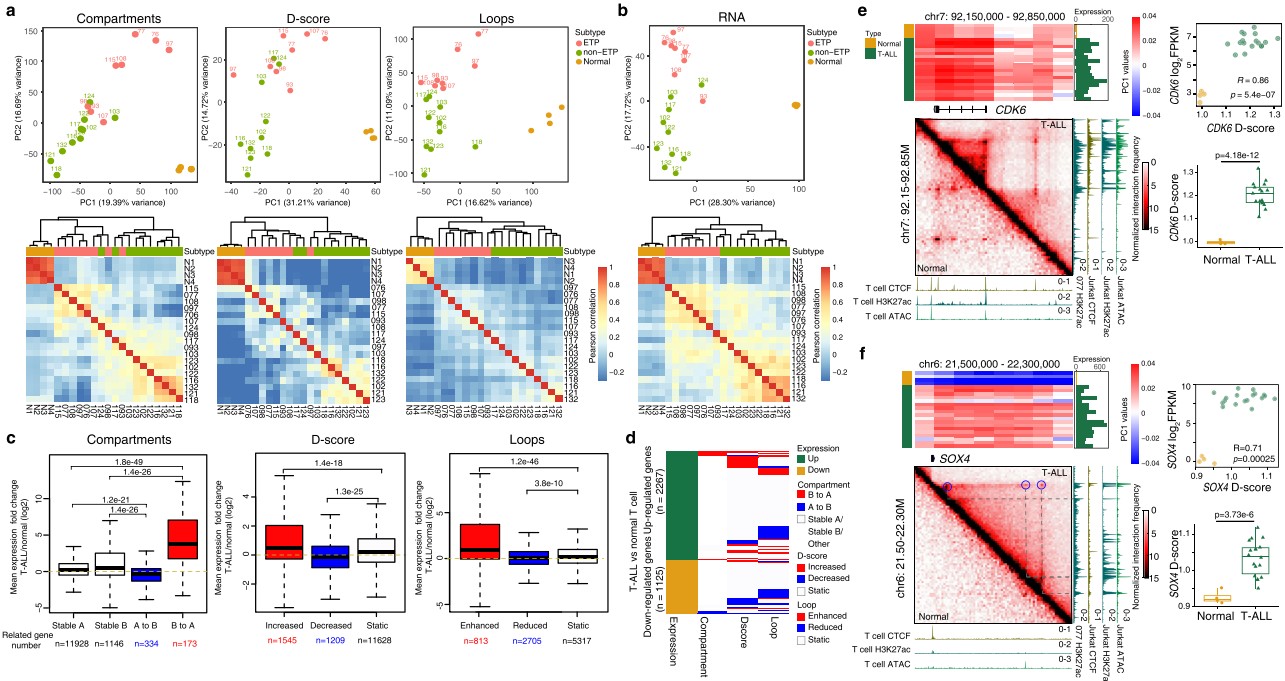

**Fig. 1 3D genome architectures in T-ALLs.** PCA (upper) and unsupervised hierarchical clustering analyses (lower) of compartment, domain score (D-score), and loop (**a**) and gene expressions (**b**) in normal T cells and T-ALLs. **c** Association of DEGs and genomic alterations at the levels of compartment, TAD, and loop. Red, upregulated; blue, downregulated. *P* values were calculated using the one-sided Wilcoxon rank sum test. **d** A summary of DEGs and their corresponding chromatin structure changes. A or B compartments were defined by the over 70% sample majority rule. "Other" refers to those states can not be defined. **e**, **f** Left-top, heatmaps show the compartment scores for each sample across the genomic regions of the *CDK6* and *SOX4* loci, respectively. Bar plots on the right show the gene expression level of *CDK6* and *SOX4* for each sample. Left-bottom, Hi-C contact maps of the *CDK6* and *SOX4* loci in normal T cell and T-ALL, each condition is normalized by its cis interaction pairs; blue circles: enhanced loops in T-ALL. ATAC-seq= and ChIP-seq tracks of CTCF and H3K27ac of normal T cells and T-ALL Jurkat cells, and H3K27ac ChIP-seq tracks of case 077 are also included. Right-upper: domain scores are plotted against gene expression of *CDK6* and *SOX4* genes. *P* values were calculated using Pearson correlation. Right-lower: quantification of the domain scores across TAD region encompassing the *CDK6* (normal T cells, n = 4 and T-ALL, n = 17. Sample 108 is excluded from **e** as it has *CDK6* related translocation) and *SOX4* genes in all samples (normal T cells, n = 4 and T-ALL, n = 18). Statistical significance was calculated using the two-sided *t*-test. In **c**, **e**, and **f**, data are represented as boxplots where the middle line is the median, the lower and upper hinges correspond to the first and third quartiles, the upper whisker extends from the hinge to the largest value no further than 1.5 × IQR from the hinge (where IQR is the inter-quartile range), and the lower whisker extends from the hinge to the smallest value at most 1.5 × IQR of the hinge, while data beyond the whiskers were ignored in **c**.

its promoter and 3 distal enhancers in T-ALL (Fig. 1f and Supplementary Data 2).

Comparing H3K27ac and CTCF ChIP data at *CDK6* and *SOX4* loci in normal T cells and T-ALL Jurkat cells, we also noticed that genomic structure changes often coincide with CTCF binding or H3K27ac modification changes. We then calculated the co-localization ratio between loop anchor and H3K27ac modification or CTCF binding sites and found that T-ALL-specific loop anchors exhibit a significant enrichment of T-ALL-specific H3K27ac modification or CTCF binding, the same pattern was also seen in normal T cells (Supplementary Fig. 1g). Together, these data suggest that 3D genome alterations as well as their associated dysregulated gene expressions are more closely associated with epigenetic changes such as CTCF binding and histone modification than CNV events.

**ETP and non-ETP ALL subtypes represent different "frozen stages" of T-cell development.** Our 3D genome landscape analyses could separate the ETP ALL samples from the non-ETP ALL samples (Fig. 1a), suggesting that the chromosomal organizations of T-ALL may represent different "frozen stages" of T-cell development[29]. To test this hypothesis, we first projected the T-ALL samples onto the T-cell developmental trajectory (Fig. 2a, upper) defined by RNA-seq analysis[30]. PCA revealed that most of the ETP ALL samples were arrested at the immature stage,

corresponding to the LMPP to Thy1 stages, while the non-ETP samples were arrested at the Thy2 to Thy4 stages (Fig. 2a, lower). The lack of TCR rearrangement in most of the ETP ALL samples and different rearrangements in individual non-ETP samples further support the notion that ETP and non-ETP ALL are arrested at different developmental stages (Fig. 2b and Supplementary Fig. 2b, c). Sample 093 was a unique case as it fell between ETP and non-ETP ALL (Figs. 1a, b and 2a) and had significant TCR rearrangement (Supplementary Fig. 2b). We also observed a lack of *RAG1* and *PTCRA* expression in most of the ETP ALL samples, which are essential for TCR V(D)J rearrangements (Fig. 2c).

Since the ETP and non-ETP ALLs can be better separated at the loop level (Fig. 1a), we further analyzed the differences in loop structures between ETP and non-ETP ALL samples and identified 1820 enhanced and 831 reduced loops in ETP ALL (Fig. 2d). When plotting gene expression changes between ETP and non-ETP ALL against the combined *p* value of the loop strength and D-score changes, we found a strong positive correlation (Pearson's correlation coefficient 0.685; Fig. 2e). Approximately 20% and 16% of the upregulated genes in ETP and non-ETP ALL, respectively, harbored consistent chromatin structure changes, including key transcription factors or oncogenes, such as *CEBPA*, *MYCN*, and *LYL1* for ETP and *LEF1*, *TCF12* and *PAX9* for non-ETP ALL (Fig. 2e and Supplementary Data 3).

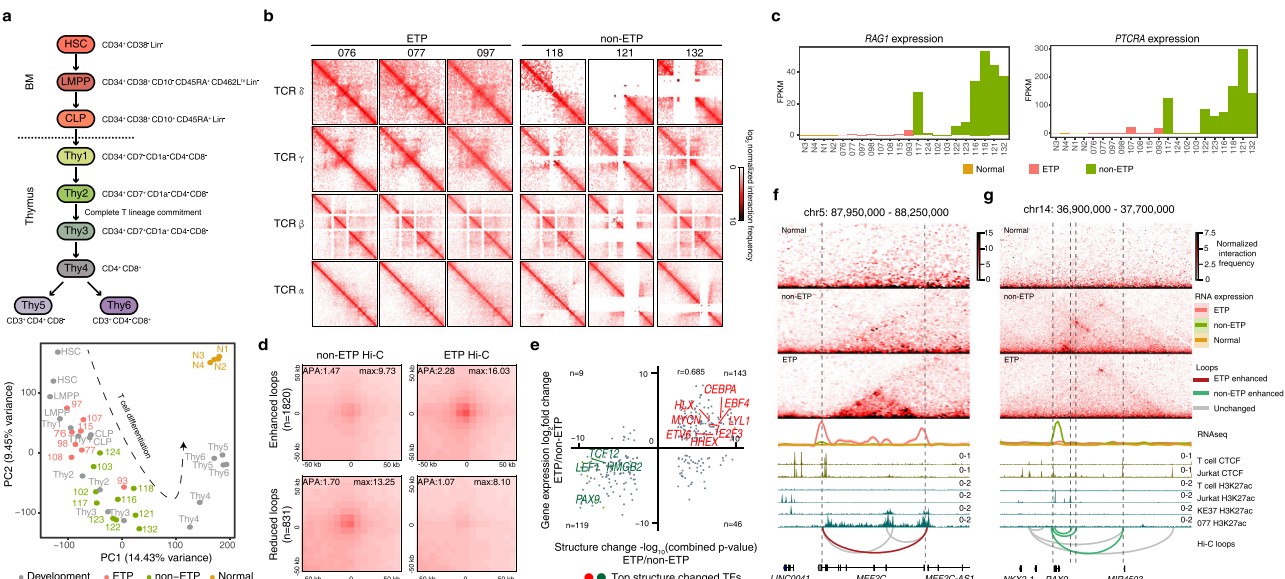

**Fig. 2 ETP and non-ETP ALLs have different loop structures. a** Upper: scheme of T-cell development of hematopoietic stem cell (HSC) and eight lymphoid cell types in human bone marrow (BM) and thymus analyzed by RNA-seq. Lower: PCA includes T-ALL cells and populations from normal human BM and thymus ($n = 2$ biological replicates per population). The trajectory of T-lineage differentiation is labeled with dash line. **b** Hi-C contact maps for the TCR genomic regions in ETP and non-ETP ALLs. **c** *RAG1* and *PTCRA* expression levels. **d** APA plots for loops that are enhanced (upper) or reduced (lower) in ETP compared with non-ETP ALL. **e** Scatterplot shows the correlation between gene expression and structural changes. The structural change was defined by the combined $p$ value of D-score and loop strength change. Top ETP and non-ETP ALL-associated transcription factors with structural changes are highlighted in red and green, respectively. Hi-C contact maps for TADs enclosing the genomic loci of ETP expressed *MEF2C* (**f**) and non-ETP expressed *PAX9* (**g**). ChIP-seq tracks of CTCF and H3K27ac of normal T cells, non-ETP ALL Jurkat cells, and ETP ALL including KE37 cells and case 077 are also included. Unchanged, non-ETP-enhanced, and ETP-enhanced loops are labeled with gray, green, and red curves, respectively. Anchors of differential loops are labeled with dashed lines.

Gene ontology analysis further revealed that genes associated with the ETP ALL-enhanced loops were enriched in immune response-activating signal transduction, myeloid cell differentiation, and regulation of B-cell activation, consistent with the definition of ETP ALL (Supplementary Fig. 2a, left). Genes associated with the non-ETP ALL-enhanced loops were enriched in terms such as positive regulation of RNA metabolism, transcription, and TCR V(D)J recombination (Supplementary Fig. 2a, right).

The 3D genome analysis also provided a potential explanation for ETP and non-ETP ALL-specific transcription factor expressions. For example, we detected subtype-specific loops and expression patterns in the *MEF2C* locus in ETP and the *PAX9* locus in the non-ETP ALL samples, respectively, which were associated with H3K27ac marks in the ETP ALL sample 077, ETP ALL cell line KE37, and non-ETP ALL cell line Jurkat (Fig. 2f, g). Collectively, the differential gene expression profiles between ETP and non-ETP ALL subtypes represent T progenitor cell arrests at different T-cell developmental stages, which are tightly associated with alterations of 3D genomes.

**Hi-C analysis revealed previous unrecognized translocations in T-ALL.** Chromosomal rearrangement is one of the major driving forces for tumorigenesis[31], especially for leukemia[1]. By adapting hic_breakfinder[32] ("Methods"), we identified 46 translocations in 14/18 T-ALL samples (Supplementary Data 4), of which 34 were newly discovered and 26 were interchromosomal events (Fig. 3a, red lines). Among 78 unique breakpoints identified, 47% located in noncoding regions, and 66% located in the stable A compartment (Supplementary Fig. 3a). These newly identified translocations not only influenced the expression of the nearest genes (Fig. 3a) but also resulted in the formation of 44 neo-loops across the translocated chromosomes, which we named

translocation-mediated neo-loops (Supplementary Fig. 3b and Supplementary Data 5, "Methods"). Interestingly, the ends of these translocation-mediated loops tend to anchor at the pre-existing loop anchors and CTCF binding sites (Fig. 3b). Importantly, nearly 78% of the translocation-mediated loops with CTCF motifs were linked to pairs of convergently orientated CTCF motifs (Fig. 3c), indicating that these loops may be mediated by loop extrusion mechanism, similar to canonical chromatin loops[15,33,34].

Clinically, non-ETP ALL can be further classified into the *HOXA*, *TLX*, and *TAL* subtypes according to their gene expression profiles[35]. Notably, there was a complete match between loop-based hierarchical clustering and non-ETP ALL subtypes, which were signified by chromosomal translocation-mediated dysregulation of T-ALL-associated transcription factors (Fig. 3d). Giving oncogenic transcription factors, such as *NOTCH*, could drive cancer-specific chromatin interactions[36], these results suggested that ectopic-expressed transcription factors may shape the subgroup-specific loop organization and cause the unique gene expression profile of each T-ALL subgroup.

**Potential mechanisms involved in translocation-mediated gene activation.** With the translocation-mediated neo-loops, we could accurately assign the possible dysregulated genes affected by the translocations. We found that translocation could potentially activate T-ALL-associated transcription factors via either "trans" or "cis" mechanism. The "trans" mechanism involves translocations within the coding regions of the genome, which mediate gene fusions, such as the *PSIP1-NUP98*, *SET-NUP214*, and *MLL*-related gene fusion events *KMT2A-MLLT1*, *PICALM-MLLT10*, and *DDX3X-MLLT10*. As reported in previous studies, these fusion events could epigenetically activate *HOXA* cluster gene expressions[37–40].

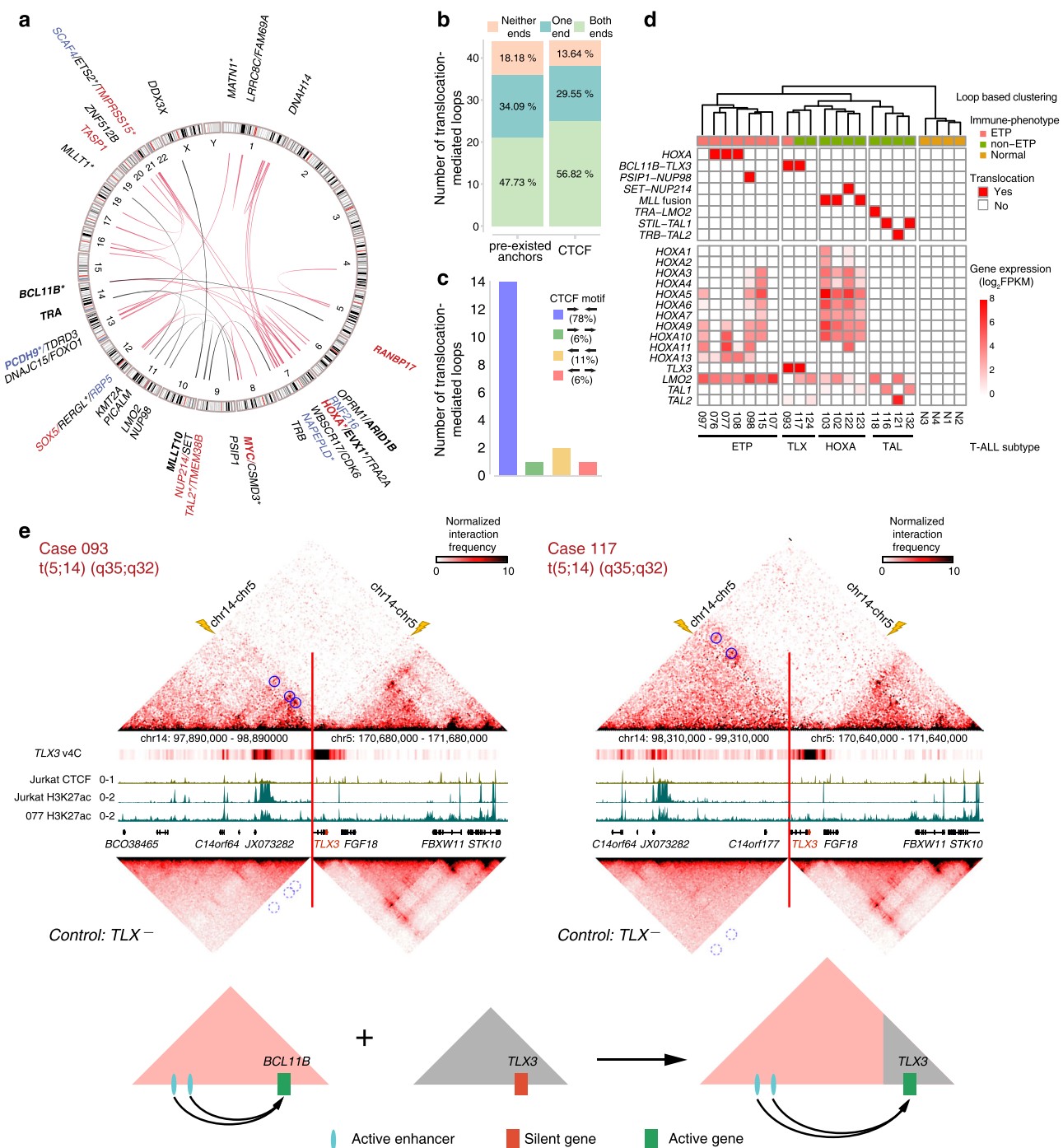

**Fig. 3 Chromosomal rearrangements in T-ALLs. a** The genomic landscape of translocations discovered by Hi-C. The previously unrecognized translocation partners are connected by red lines and known translocations are shown by black lines. Breakpoints nearest genes with increased or decreased expressions are highlighted by red and blue, respectively. Recurrent breakpoints nearest genes are labeled in bold. The genes closest to the breakpoint in noncoding regions are marked by star symbols. **b** Number and percentage of translocation-mediated loops with anchors overlapped by pre-existed anchors (normal loop anchors) or CTCF peaks. **c** Number and percentage of translocation-mediated loops with different CTCF motif orientation. **d** The loop-based clustering overlaps with leukemogenic transcription factor- or translocations-based clustering in T-ALLs. *STIL-TAL1* fusions were detected by RT-qPCR, the other translocations were detected by Hi-C. **e** Upper panels: Hi-C heatmaps for cases with *BCL11B-TLX3* translocation (top), visual 4C plots generated from Hi-C contact maps using *TLX3* promoter as the viewpoint (middle) and averaged Hi-C heatmaps for cases without *BCL11B-TLX3* translocation as controls (bottom). ChIP-seq tracks of CTCF and H3K27ac of Jurkat cells and case 077 are also included. Breakpoints are marked by the red lines and yellow lightning bolts. Translocation-mediated loops are highlighted by blue circles, corresponding loop locations at controls are highlighted by dotted line circles. Lower panels: a schematic illustration for the consequence of the *BCL11B-TLX3* translocation.

The "cis" mechanism involved translocations within the noncoding region of the genome in the ETP, *TLX*, and *TAL* subtypes (Fig. 3d and Supplementary Fig. 3b), of which the ectopically expressed transcription factors, such as *TLX3*, could hijack the enhancers from the translocated *BCL11B* via translocation-mediated neo-loop formation (Fig. 3e and Supplementary Fig. 3c, d). This "cis" mechanism could involve 2 *BCL11B-TLX3*, 1 *TRB-TAL2*, and 3 *HOXA13* translocations identified in this study (Supplementary Fig. 3b and Supplementary Data 5). Interestingly, most of these putative hijacked enhancers are from genes that are normally expressed during T-cell development, such as *BCL11B* and *TRB*, which could lead to ectopic expression of T-ALL-associated transcription factors in the T lineage and block normal differentiation (Supplementary Fig. 3d).

**3D genome alterations and ectopic HOXA gene expressions**. The dysregulated *HOXA* cluster is a common feature of T-ALL[35] and associated with poor prognosis[41,42], we therefore focused our attention on the relationship between 3D genome alterations and ectopic *HOXA* gene expressions. We conducted unsupervised hierarchical clustering based on the levels of *HOXA* gene expressions, which separated 15 T-ALL samples without *HOXA* translocation into *HOXA*-negative (*HOXA⁻*) and *HOXA*-positive expression (*HOXA⁺*) groups. Translocation-mediated fusion events could be detected in 5/7 *HOXA⁺* samples, implying that translocation may be the major driving force for *HOXA* activation. The *HOXA⁺* T-ALLs could be further separated into two subgroups: the 3′*HOXA⁺* or 5′*HOXA⁺* subgroups, with respect to the location of the *HOXA* genes within the *HOXA* cluster (Fig. 4a, b). The expression patterns of the three *HOXA13* translocation cases (*HOXA13*-T, breakpoints shown in Fig. 4b) were closer to those in the 5′*HOXA⁺* subgroup, characterized by ectopic *HOXA13* expressions (Fig. 4a).

The *HOXA* cluster, which contains 11 genes, is transcriptionally repressed in normal T cells but can be transactivated in T-ALLs by fusion proteins that recruit histone methyltransferase DOT1L to the *HOXA* locus[37,43]. Although this mechanism uncovered how the *HOXA* cluster is activated, it cannot explain the diverse *HOXA* expression patterns associated with different fusion proteins. By integrating Hi-C maps with *HOXA* gene expression patterns, we found that the differential *HOXA* gene expressions were associated with different 3D genome organizations.

Hi-C maps and CTCF motif orientations showed that the 11 *HOXA* genes were partitioned between two TADs (Fig. 4b and Supplementary Fig. 4a): the CTCF binding site C11/13 was used as the 3′ boundary of the 5′ TAD in all samples, while the 5′ boundaries of the 3′ TADs varied among different samples: C7/9 was used by most of the *HOXA⁻* (6/8) and all 5′*HOXA⁺* samples (2/2), while C10/11 was used by most of the 3′*HOXA⁺* samples (4/5) (red and blue arrows/lines, respectively; Fig. 4b and Supplementary Fig. 4a).

We further identified six enhancer regions in each of the TAD, labeled as E1–E12, which could interact with the *HOXA* cluster (Fig. 4c). Using *HOXA⁻* cases as common denominators (Fig. 4c, top panel), we calculated the overall differential interaction intensities. Although there was no significant difference between the healthy controls and the *HOXA⁻* cases in the 12 interaction regions, we found significantly enhanced interactions between E2–E6 and genes in the 3′*HOXA⁺* subgroup, as well as between E8, 9, 11, 12 and genes in the 5′*HOXA⁺* subgroup, either as a group average (Fig. 4c) or individually (Supplementary Fig. 4b, c). ChIP-seq analysis of the *HOXA⁺* Loucy cell line indicated that these increased interactions may be correlated with gains in the H3K27ac histone mark (Fig. 4c). These results suggest that 3D genome organization is closely associated with the patterns of ectopic gene expression within the *HOXA* cluster.

**Unique fusion events associated with ectopic HOXA11-A13 gene expression**. By investigating the associations between different fusion events and ectopic expression of the *HOXA* genes, we found that *MLL*-related fusion events, such as *KMT2A-MLLT1, PICALM-MLLT10*, and *DDX3X-MLLT10*, were associated with the 3′*HOXA* subgroup, the *PSIP1-NUP98* fusion event was associated with the 5′*HOXA* subgroup, while *SET-NUP214* fusion events were associated with both subgroups (Fig. 4d).

We further verified these results by including a larger cohort of *HOXA⁺* T-ALL samples from St. Jude data set[6]. Out of 131 *HOXA*-positive T-ALL samples (82 from St. Jude, 39 from in-house data, and 10 from our study), we found 41 with aforementioned fusion events and 4 with *HOXA13*-T. Among *MLL*-related fusions, 29/32 had *HOXA7* expression, a signature for 3′*HOXA* subgroup, while 5/6 *NUP98*-related fusions were *HOXA13*-positive, a signature for 5′*HOXA* subgroup. *SET-NUP214* fusion samples showed no specificity (Fig.4e and Supplementary Fig. 4d). These results suggest that different translocation-mediated fusion events may preferentially alter the genome interactome of the *HOXA* cluster and control a specific set of *HOXA* gene expression.

**Activation of 5′HOXA genes via potential enhancer hijacking**. To explain the gene expression patterns seen in the four *HOXA13*-T samples, we mapped the breakpoints of three T-ALL samples with Hi-C data. We found that all the breakpoints located within the 5′ TAD, upstream of the *HOXA13* gene. The translocation partner breakpoints lie in the gene bodies of the *BCL11B* and *CDK6* genes on chromosomes 14 and 7, respectively, as well as upstream of the *ERG* gene on chromosome 21 (Fig. 5 and Supplementary Fig. 5). By examining the TAD structures associated with the translocations, using *HOXA⁻* samples as controls, we found translocation-mediated neo-loop formations between the 5′ of the *HOXA* cluster and cis-regulatory elements associated with *BCL11B, ERG*, and *CDK6* genes (Fig. 5a–c). *BCL11B, ERG*, and *CDK6* were expressed in these T-ALL samples (Supplementary Figs. 4c and 5a, b), and ChIP-seq analysis of blast cells from case 077, Jurkat and Loucy cells did confirm that the elements associated with the neo-loops were marked by H3K27ac, a sign for active enhancers (Fig. 5a–c). These results suggest that the active enhancers from the *BCL11B, ERG*, and *CDK6* genes juxtapose to upstream *HOXA13* after translocation, promoting ectopic expression of *HOXA13* in the case of 076 and *HOXA9-A13* in the case of 077 (blue circles for neo-loops and green bar graphs for gene expressions; Fig. 5a, b and Supplementary Fig. 3b). For the case of 108, the inter-TAD inversion could potentially lead to the adoption of the active CDK6 enhancers and neo-loop formation, leading ectopic expression of the *HOXA11-A13* genes (Fig. 5c).

To directly assess the capacities of the putative enhancers to potentiate gene expressions, we cloned DNA fragments corresponding to the putative enhancers or adjacent DNA fragments into the luciferase reporter construct (E, enhancer fragment; C, adjacent control fragment; Fig. 5a–c and Supplementary Fig. 5c–e). When expressed in Jurkat cells, these putative enhancers led to robust reporter activities as compared to the control fragments (Fig. 5d). Most of the putative enhancers also showed robust reporter activities in Loucy cells (Supplementary Fig. 5f). These results support the notion that translocation and inversion could bring active enhancers close to otherwise silenced 5′*HOXA* genes through enhancer hijacking, leading to dysregulated gene expressions. The existence of the CTCF binding sites

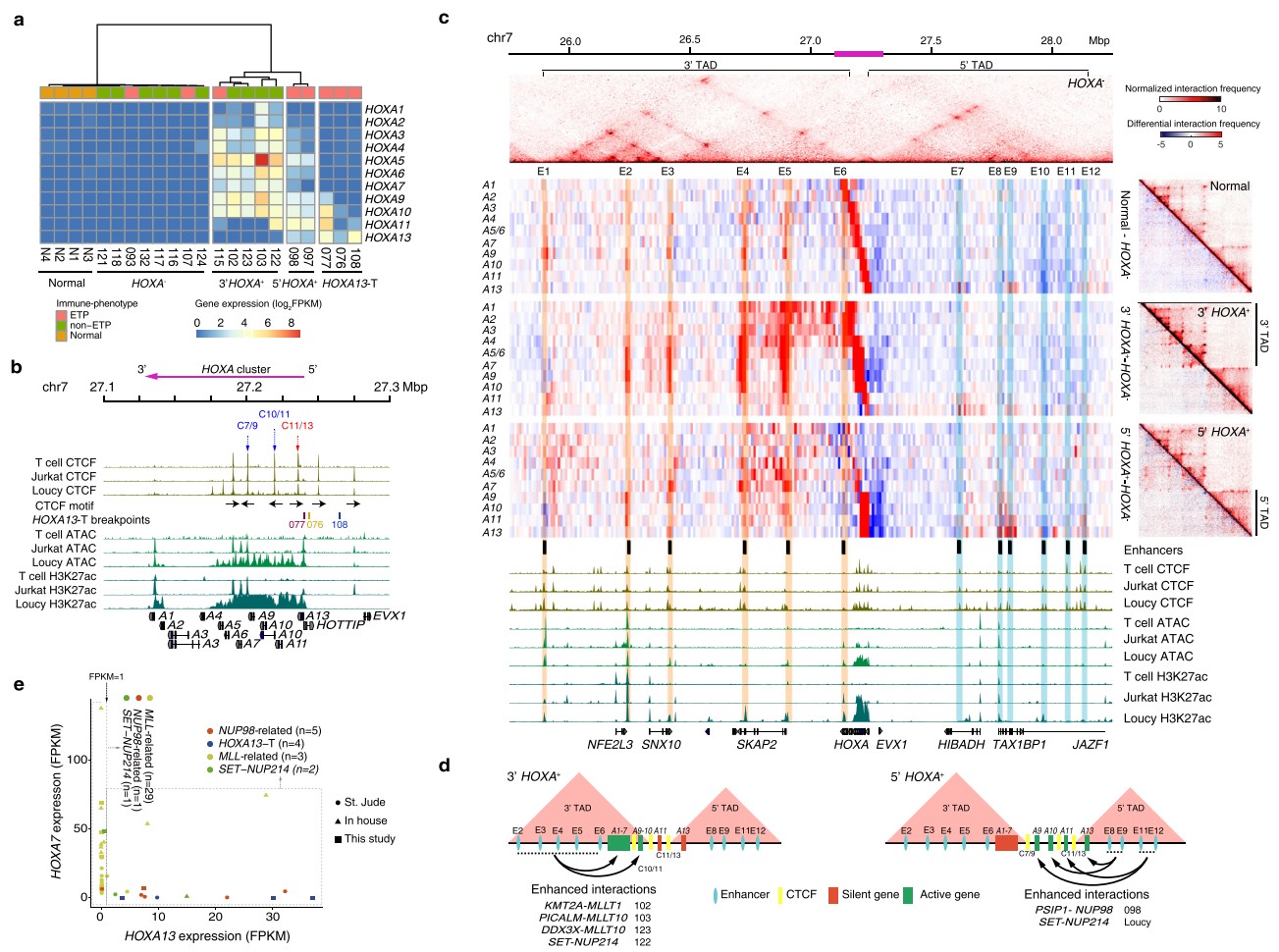

**Fig. 4 Chromatin interaction profile and expression patterns of the *HOXA* cluster in T-ALLs. a** T-ALL subtypes with different *HOXA* gene expression patterns. *HOXA13*-translocation (*HOXA13*-T) negative cases are grouped by unsupervised hierarchical clustering based on the *HOXA* gene expressions. **b** ChIP-seq tracks of CTCF and H3K27ac and ATAC-seq tracks in T cell, *HOXA*⁻ Jurkat and *HOX*A⁺ Loucy T-ALL cells in the genomic region corresponding to the pink bar of Fig. 4c. CTCF motif orientations, 5′TAD and 3′TAD boundaries and *HOXA*13-T breakpoints are also included. **c** Top: a Hi-C heatmap shows the average interaction intensity of *HOXA*⁻ cases in chr7: 25,750,000-28,250,000 (hg19), which includes *HOXA* gene cluster and its 3′ and 5′ TADs. Middle left: from top to bottom, Hi-C heatmaps show differential interaction intensities between normal T cell vs. *HOXA*⁻, 3′ *HOXA*⁺ vs. *HOXA*⁻ and 5′ *HOXA*⁺ vs. *HOXA*⁻ cases using visual 4C plot. Each line represents interactome of a *HOXA* gene shown on the left. *HOXA5* and *HOXA6* share one line as they located in the same bin. The main enhancers are highlighted with orange in the 3′ TAD and blue in the 5′ TAD. Middle right: top right of each heatmap represents the averaged interaction intensities of normal T cell, 3′ *HOXA*⁺ cases and 5′ *HOXA*⁺ cases, respectively; bottom left of each heatmap represents the differential interaction intensities between each group and *HOXA*⁻ cases. Bottom: ChIP-seq tracks for CTCF and H3K27ac and ATAC-seq tracks in T cell, *HOXA*⁻ Jurkat and *HOXA*⁺ Loucy T-ALL cells. **d** Schematic illustrations of the associations among 3D-genomic interactions, *HOXA* cluster expressions, and associated fusion events in 3′ *HOXA* cases (left) and 5′ *HOXA* cases (right). **e** *HOXA13* and *HOXA7* expression of *HOXA*-positive samples with *MLL*-related (n = 32), *NUP98*-related (n = 6), *SET-NUP214* fusions (n = 3), and *HOXA13*-T (n = 4). Samples are colored according to translocation.

C11/13 in the case of 076, C7/9 in the case of 077, and C10/11 in the case of 108 (Fig. 5e) may insulate the activate enhancer spreading to *HOXA* genes located in the 3′ TAD, leading to 5′ *HOXA*-specific expression patterns (Fig. 4b and Supplementary Fig. 4d).

### *HOXA11-A13* positivity is linked to inferior outcome in pediatric T-ALL. With heterogenous *HOXA* gene expression patterns and different activation mechanisms discovered by Hi-C, we next investigated whether *HOXA*⁺ cases also represent heterogeneous clinical entities. By analyzing a cohort of T-ALL patients with outcome information (our unpublished results), we found that *HOXA13* or *HOXA11* positiveness, alone or in combination, but not the expression of other *HOXA* genes, such as the previously used biomarker *HOXA9*, was associated with poor overall and event-free survivals in young adult and pediatric T-ALLs (Fig. 6a and Supplementary Fig. 6a, b). In multivariate

analysis, *HOXA13*⁺ status could serve as an independent predictor for the overall survival of pediatric and young adult T-ALLs (Fig. 6b).

As almost all cases with *HOXA13*-T and *NUP98*-related fusion events belong to the ETP ALL subtype and were associated with ectopic expression of the *HOXA11-A13* genes (Fig. 4a and Supplementary Fig. 4d), we extended our analysis by integrating our in-house data with data published by Liu et al.[6] to further characterize the *HOXA11-A13*⁺ group. Compared to the *HOXA1-A10*⁺ cases, *HOXA11-A13*⁺ cases were indeed enriched for ETP ALL subtype in this large cohort of 273 patients (Fig. 6c). We also checked mutations associated with the *HOXA11-A13*⁺ group in 325 samples with WES data and found that the rate of JAK-STAT pathway-associated mutations was significantly higher in the *HOXA11-A13*⁺ group than that of the *HOXA1-A10*⁺ group (Fig. 6d). Therefore, *HOXA11-A13* positiveness, which can be identified either by gene expression or cytogenetic

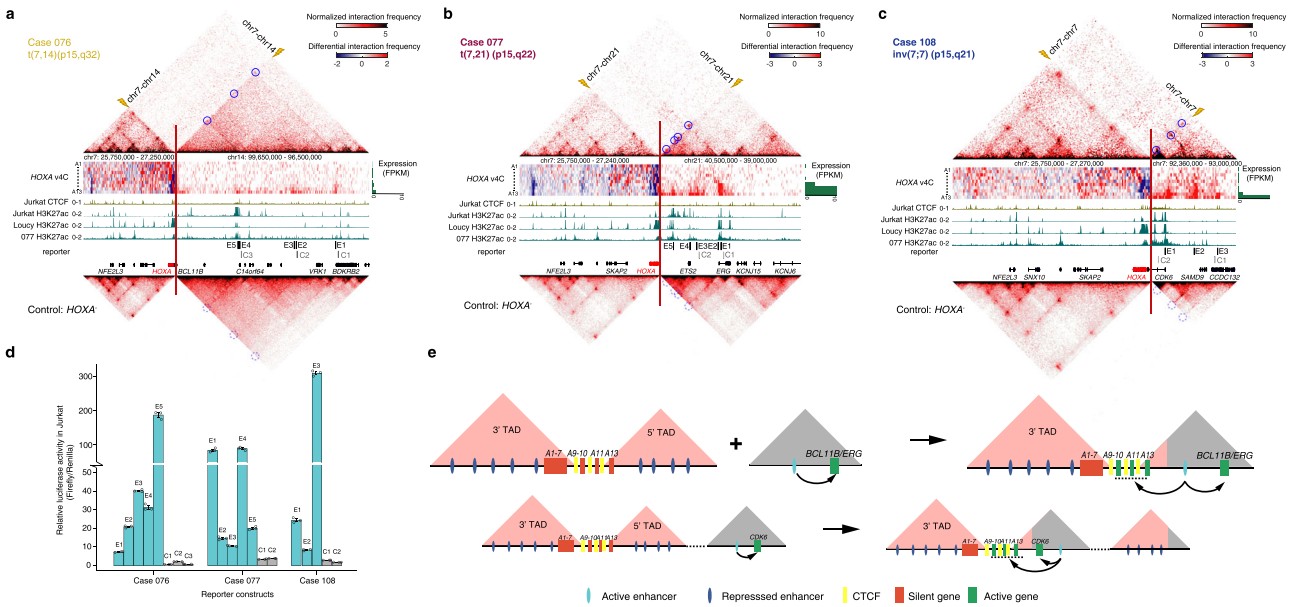

**Fig. 5 Translocation-mediated enhancer hijacking and ectopic *HOXA* gene expressions in T-ALLs. a–c** Individual Hi-C heatmaps for three *HOXA13*-T cases (top) and averaged Hi-C heatmaps of *HOXA*⁻ control cases (bottom). Middle panels are differential visual 4C plots between *HOXA13*-T cases and *HOXA*⁻ using *HOXA1-A13* genes (from top to bottom) as the viewpoint. ChIP-seq tracks of CTCF of Jurkat and H3K27ac of Jurkat, Loucy, and case 077 are also included. The green bar graphs on the right show the expression level of each *HOXA* gene. Breakpoints are marked by the red lines and yellow lightning bolts. Translocation-mediated loops are highlighted by blue circles, corresponding locations in control samples are marked by dotted circles. Location of predicted enhancers and control regions are marked by black and gray bars, respectively. **d** Luciferase reporter activity for regions encompassed within the predicted enhancers and control regions indicated in **a–c** compared to empty vector in Jurkat cell line (*n* = 3 per group). Data are represented as mean ± SD. **e** Schematic illustrations of the consequences of *HOXA13*-related translocations (upper) and inversion (lower).

analysis of translocation and fusion events identified in this study, may serve as a biomarker for identification of T-ALL patients with poor prognosis. Anti-JAK-STAT inhibitor treatment may also benefit this group of patients.

## Discussion

Most studies on the underlying mechanisms of T-ALL leukemogenesis are focused on the coding regions of the genome and many disease-driving genes and pathways have been identified through WES or RNA-seq analyses. Our work, together with recent publication by Kloetgen et al.[18], provides a comprehensive view of the 3D chromosomal structures of T-ALL. Both works demonstrate that global 3D genome architecture can separate normal T cells and two T-ALL subtypes (Fig. 1), although precaution must be taken in interpreting this result as the differences detected by Hi-C may simply reflect the different developmental stages corresponding to T-ALL and normal T cells. The chromosomal structure-based clustering is consistent with gene expression-based grouping, implying that 3D genome alterations may be responsible for dysregulated gene expressions in T-ALL (Fig. 2). Indeed, in our study ~29% differentially expressed genes between T-ALL and normal T cell, including those dysregulated key transcription factors in ETP and non-ETP ALL, are associated with 3D genome alterations but not CNVs. While Kloetgen et al.[18] focused on NOTCH pathway regulated genomic structures and recurrent TAD fusion events that regulating *MYC* expressions, we focused our attention on the translocation events previous unknown to the field, especially those translocations within the noncoding regions of the genome. Overall, both studies suggest that 3D genome alterations may be contributing factors for T-ALL leukemogenesis.

By employing high-resolution Hi-C map, we identified 34 translocations in T-ALL that cannot be recognized by RNA-seq and WES (Fig. 3). Dysregulated oncogenic transcription factors,

such as TAL1, TAL2, TLX3, LMO2, and HOXA were associated with translocation-mediated genomic alterations in 78% (14/18) cases, suggesting that translocations may play driver roles in T-ALL leukemogenesis. We further provided detailed contacting maps between the *HOXA* cluster and the two TADs surrounding the cluster (Fig. 4). We demonstrated the tight associations among enhanced 3′TAD contacts, *MLL*-related gene fusion events, and 3′*HOXA* gene expressions. Similarly, the 5′*HOXA* gene expression pattern, enhanced 5′TAD contacts, and *NUP98*-related fusion events showed positive correlation. These results suggest that different fusion events may differentially activate *HOXA* gene expressions by enhancing the interaction between 3′- or 5′-TAD and the *HOXA* cluster. Importantly, we demonstrated translocation-mediated "neo-TAD" and "neo-loop" formation on the Hi-C contact maps of three *HOXA13* translocation cases, and potential enhancer-hijacking mechanisms that were likely involved in T-ALL development (Fig. 5). Further study is required to demonstrate the causality between the "neo-TAD" and "neo-loop" and dysregulated transcription factor.

Our 3D structure-based analysis helped to separate T-ALL patients with ectopic *HOXA* cluster expressions into two groups, one with *HOXA11-A13* expressions and the other with *HOXA1-10* expressions (Figs. 4 and 6). Although ectopic expressions of the *HOXA1-10* genes did not contribute to the survival of T-ALL patients, *HOXA11-A13* expressions could serve as an independent predictor for poor overall survival of pediatric and young adult T-ALLs. Patients with ectopic expressions of *HOXA11-A13* genes were associated with *HOXA13* translocation and *NUP98*-related fusions, which could be readily identified by cytogenetic analysis. Patients with ectopic expressions of *HOXA11-A13* genes also have higher rate of JAK-STAT pathway mutations, suggesting that anti-JAK-STAT inhibitor treatment may benefit this group of patients. Together, our study indicates that 3D genome alterations may contribute to T-ALL development by regulating adjacent or distant

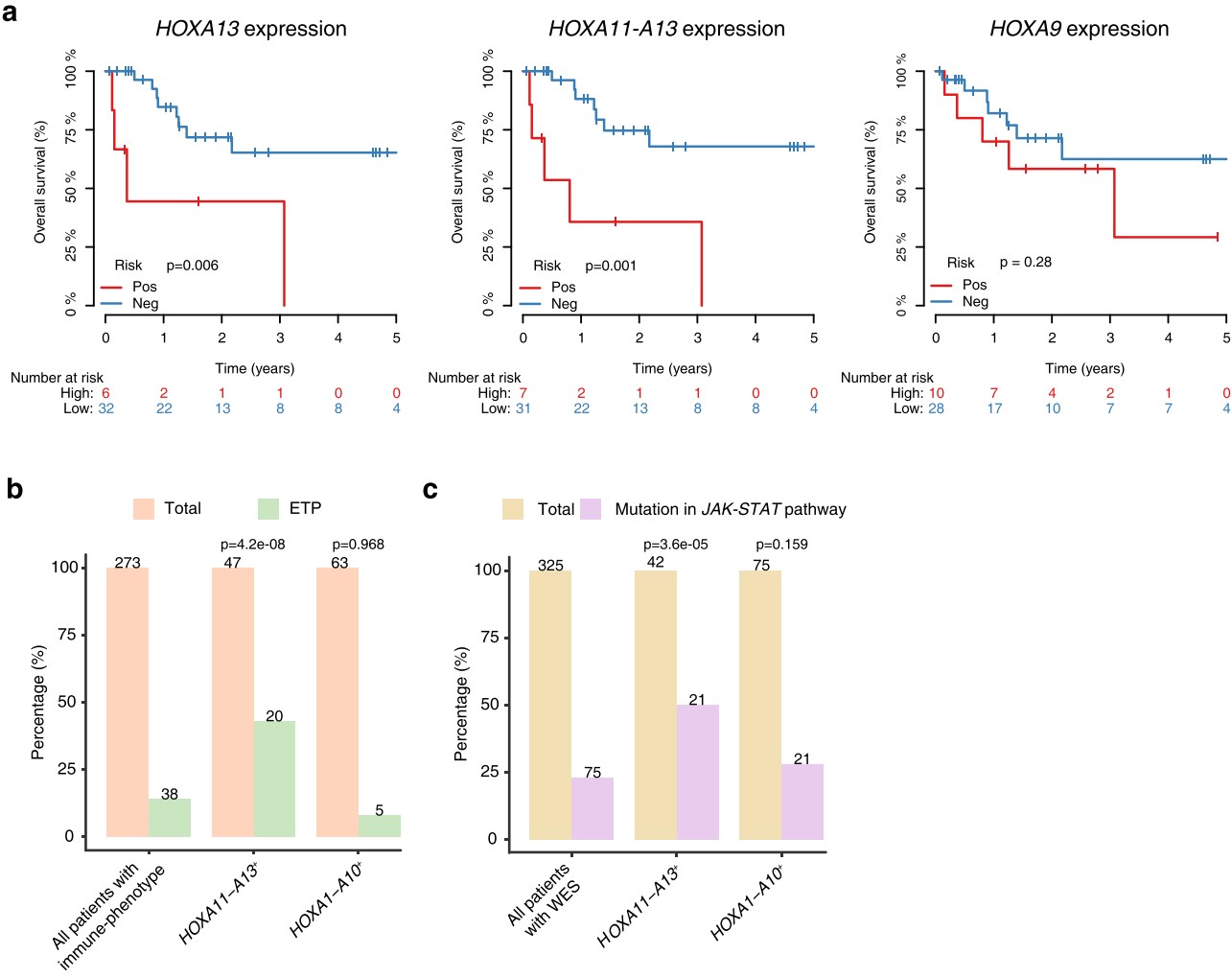

**Fig. 6 Ectopic *HOXA11-A13* expressions are correlated with poor outcomes in pediatric and young adult T-ALLs. a** Kaplan–Meier overall survival curves for patients with (red) and without (blue) ectopic *HOXA13* expression (left), *HOXA11-A13* expressions (middle) and *HOXA9* expressions (right) in pediatric and young adult patients. *P* values are based on the two-sided log-rank test. **b** The associations between ectopic *HOXA11-A13* expression and ETP phenotypes. *P* values are calculated by the one-sided Fisher exact test. **c** The proportion of cases with JAK-STAT pathway mutations in each T-ALL subgroup, *P* values are calculated by the one-sided Fisher exact test.

gene expressions via "cis" or "trans" mechanism and understanding the underlying mechanism may inform new strategies for patient stratification and targeted therapy.

## Methods

**Patients and samples**. The T-ALL samples were collected from diagnostic bone marrow aspirations at the Peking University People's Hospital of China. The patient characteristics are described in Supplementary Data 6. ETP ALL status was defined as previously published[3]. Leukemia blast cells were prepared by density-gradient centrifugation of bone marrow samples, and CD19−CD14−CD235−CD45+CD7+ cells were further purified by flow cytometry sorting using antihuman antibodies for RNA-seq and Hi-C library preparations. Peripheral blood samples were obtained from four healthy donors. T cells were purified using the EasySep™ Direct Human T Cell Isolation Kit (StemCell Technology #19661). This study was approved by the Ethics Committee of Peking University People's Hospital. All patients and healthy donors provided written informed consent before any study procedure.

**RNA-seq library preparation, data processing, and differential gene expression analysis**. RNA-seq libraries were prepared with the TruSeq RNA Library Prep Kit v2 (Illumina). Paired-end RNA-seq reads of the 18 patients and 4 healthy controls were generated with an average depth of 15 million read pairs. Reads were aligned to the hg19 genome with TopHat (v2.1.0) using default settings[44]. Duplicates were removed, and aligned reads were calculated for each protein-coding gene using HTSeq[45], followed by FPKM transformation by normalizing gene exon

length and sequencing depth. Raw RNA-seq data for Loucy and Jurkat were downloaded from the GEO database and analyzed as described above.

DESeq2[46] was applied to identify the differentially expressed genes with FDR < 0.01 and fold change >2. Genes with fewer than five reads in 20% of the samples or with mean reads fewer than two were excluded. Signal tracks were generated by using BEDTools[47] genomeCoveragebed to produce bedGraph files scaled to one million reads per data set. Then, the UCSC Genome Browser utility[48] bedGraphToBigWig was used with default parameters to generate bigwig files.

**ChIP-seq, ATAC-seq data processing, and motif analysis**. H3K27ac ChIP of case 077 was performed on 10^5 cells, according to the protocol of Hyperactive In-Situ ChIP Library Prep Kit for Illumina (Vazyme, TD 901). ChIP-seq reads were mapped to the hg19 genome with Bowtie2[49] (v2.3.5) using default parameters, while ATAC-seq reads were mapped with Bowtie2 using parameter -X 2000 λno-mixed --no-discordant --no-unal. Aligned reads were filtered for a minimum MAPQ of 20, and duplicates were removed using SAMtools[50]. Signal tracks and peaks were generated by using the –SPMR option in MACS2[51]. Then, the UCSC Genome Browser utility bedGraphToBigWig was used with default parameters to transform the bedgraph files to bigwig files. FIMO[52] was used to detect the 20-bp CTCF motif from the Homer motif database in Loucy and Jurkat CTCF peaks with default parameters.

**Hi-C and Hi-C data processing**. Hi-C was performed on one million cells/sample, according to the BL-Hi-C protocol[27]. Raw BL-Hi-C reads were processed by the in-house HiCpipe framework, which integrated several Hi-C analysis methods to generate multiple features of the Hi-C data. In particular, ChIA-PET2[53] was used

to trim the bridge linkers and HiC-Pro[54] to align reads, filter artifact fragments, and remove duplicates; Juicer[9] was applied to the resulting uniquely mapped contacts to generate individual or merged Hi-C files that could be deposited as contact matrices with multiple resolutions. Knight-Ruiz[55] (KR)-normalized matrices were used in the compartment and TAD analyses.

**Compartment and TAD analysis.** The compartment was calculated with the eigenvector command of Juicer under 100-kb resolution KR normalized Hi-C matrices. For every 100-kb bin, A or B compartments were defined by the over 70% sample majority rule.

TAD boundaries were calculated by the Insulation score method[56] (with parameters: -is 1000000 -ids 200000 -im mean -nt 0.1) on pooled 40-kb Hi-C matrices of the healthy T-cell controls, ETP, and non-ETP samples. The resulting TAD boundaries were merged and assigned with relative insulation scores of all samples calculated from HiCDB[57]. Differential TAD boundaries were defined with a t-test FDR < 0.01 and a difference between cases and controls higher than 50% quantile of the overall difference.

A TAD was defined when its boundaries were detected in at least two conditions among normal T cells, ETPs, and non-ETPs. The domain score[58] was calculated in each sample by dividing the intra-TAD interactions with all interactions connected to the corresponding TAD. Differential domain scores were calculated with a t-test FDR < 0.01 and fold change higher than 70% quantile of the overall fold change.

**Loop detection and differential loop calling.** Loops were called by HiCCUPS[9] at 5- and 10-kb resolutions with default parameters (except -d 15000,20000) for pooled Hi-C matrices of the healthy T-cell controls, ETP and non-ETP samples, respectively. The differential loop detection method was adapted from Phanstiel et al.[59]. Loops were split into two distance ranges (> or <150 kb) to minimize potential bias (Rubin et al.[60]). Differential loops were called within each range (FDR < 0.1) and then combined.

**Loop aggregation and functional analysis.** Aggregate peak analysis (APA) plots were generated to assess the quality of loop detection and explore the characteristics of different loop classes by the Juicer APA command[9] under 5-kb resolution. Its output matrix was normalized by the loop number that contributed to the matrix generation. For analysis of the function of dynamic loops between non-ETP and ETP, the loop anchors were analyzed by GREAT[61] (v3.0.0) using the nearest gene within 100 kb to generate the enriched biological process. In other sections, genes related to loops were determined if their promoter (5 kb around TSS) overlapped the loop anchors, and DAVID[62] 6.8 was used for KEGG pathway enrichment analysis.

**Visualization and V4C plot generation.** Tracks of Hi-C maps and ChIP-seq were generated by pyGenomeTracks[63]. Hi-C maps of each condition were normalized by its cis interaction pairs. A visual 4C (V4C) plot for specific loci was generated as the interactions related to the corresponding viewpoint under 10-kb resolution.

**Translocation and translocation-mediated loop detection with Hi-C.** hic_breakfinder[32] was adapted to detect translocations in 18 T-ALL patient

samples. After we filtered the "translocations" also detected in normal controls, the remaining translocations were manually assessed, and the precise breakpoints were determined. As the average depth of patient Hi-C samples is 486 million read pairs and the read length is 150 bp, Hi-C raw data were treated as single ends to refine the breakpoint locations to single base-pair resolution. Any single ends that could be mapped to two different chromatins without the BL-Hi-C bridge linker in between were chimeric reads. The chimeric reads detected from the BL-Hi-C data overlapped with the aforementioned translocations at 5–20-kb resolution. For each translocation, the exact locations (single base-pair resolution) supported by more than three chimeric reads were identified as the actual breakpoints and are reported in Supplementary Data 4.

As translocation-mediated loops were hard to identify by loop detection tools designed for intrachromosomal loop detection and easy to capture by visualization, their locations were manually recorded on interchromosome Hi-C maps with the help of Juicebox[64], which is an interactive visualization software.

**Translocation, translocation-mediated loop annotation and visualization.** The nearest genes to translocation breakpoints were determined by BEDTools. Known translocations were collected from refs. [6,8] and ChimerPub[65]. A translocation was considered novel if any of the breakpoints was not near any known breakpoint within a 100-kb distance. For translocation-mediated loops, the genes with a promoter or gene body overlapping the loop anchors were annotated as the associated genes. A gene near a breakpoint was considered upregulated if its FPKM was greater than one- and twofold higher than the control samples without nearby breakpoints. Hi-C heatmaps and Visual 4C plots of the reassembled chromatin were generated by MATLAB code.

**Luciferase enhancer assays.** Candidate enhancer and negative control regions were amplified by PCR using the primer sets listed in Supplementary Table 1 and cloned into the pGL4.26 vector (Promega) containing a multiple cloning region for insertion of a response element of interest upstream of a minimal promoter and the firefly luciferase reporter gene. Firefly luciferase constructs and control reporter Renilla luciferase vector were co-electroporated Jurkat and Loucy cells with a Celetrix Transfection System Device (Dakewe, CTX-1500A). Electroporation conditions: Vset = 440 V, Tset = 30 ms, Punm = 1n, Tint = 1 ms. Luciferase activity was measured 60 h after electroporation with Dual-Luciferase® Reporter Assay System (Promega, E1960) by Glomax® 20/20 Lunimometer (Promega).

**Statistics.** Specific statistical analyses are described in each section. In general, the Wilcoxon rank sum test was employed in R for comparisons of distributions. Survival analysis was performed by a Cox regression model using overall and event-free survival as outcomes. Overall survival was defined as the time from diagnosis to death from any cause. Event-free survival was defined as the time from diagnosis to treatment failure, relapse, or death from any cause. The proportional hazard assumption was tested. Variables tested in the multivariable Cox regression model were sex, age (pediatric vs. adult), white blood cell counts, hemoglobin levels, platelet counts, hepatosplenomegaly, percentage of blasts in the bone marrow, and MRD status. For in-house data, 86 patient samples with RNA-seq data were used for ETP enrichment analyses, of which 38 samples under the age of 40 with outcome and 63 samples with whole-exon sequencing (WES) data were used for survival and mutation analyses, respectively.

**Table 1 Univariable and multivariable analysis of overall survival according to *HOXA13* or *HOXA11-13* expression and select variables.**

| | Univariate | | | Multivariate (HOXA13) | | | Multivariate (HOXA11-13) | | |
|---|---|---|---|---|---|---|---|---|---|
| | HR | 95% CI | P value | HR | 95% CI | P value | HR | 95% CI | P value |
| HOXA13 expression (FPKM) ≥1 vs. <1 | 4.675 | 1.39–15.69 | 0.013 | 38.25 | 1.37–1071 | 0.032 | | | |
| HOXA11-13 expression (FPKM) ≥1 vs. <1 | 5.872 | 1.84–18.74 | 0.003 | | | | 8.84 | 0.82–95.24 | 0.072 |
| Gender Males vs. female | 1.13 | 0.42–3.01 | 0.806 | 0.959 | 0.05–17.78 | 0.978 | 0.89 | 0.066–12.015 | 0.93 |
| Age Pediatric vs. adult | 0.21 | 0.07–0.64 | 0.006 | 0.154 | 0.006–3.684 | 0.248 | 0.34 | 0.014–8.444 | 0.512 |
| WBC Low vs. high | 1.46 | 0.62–3.43 | 0.381 | 73.52 | 1.59–3390 | 0.028 | 31.95 | 1.5–680.5 | 0.0264 |
| Hemoglobin | 1 | 0.99–1.02 | 0.571 | 1.02 | 0.94–1.11 | 0.574 | 1.008 | 0.94–1.077 | 0.813 |
| Platelet | 1 | 0.99–1 | 0.952 | 0.986 | 0.968–1.004 | 0.127 | 0.987 | 0.971–1.004 | 0.139 |
| Hepatosplenomegaly Yes vs. no | 1.47 | 0.34–6.33 | 0.605 | 0.24 | 0.004–15.882 | 0.504 | 1.67 | 0.022–128.414 | 0.817 |
| Blasts in BM | 1.01 | 0.98–1.03 | 0.621 | 1.107 | 0.906–1.352 | 0.319 | 1.08 | 0.9–1.294 | 0.407 |

*WBC* white blood cell count, *HR* hazard ratio, *CI* confidence interval.

**Reporting summary**. Further information on research design is available in the Nature Research Reporting Summary linked to this article.

## Data availability
The raw sequence data of Hi-C, RNA-Seq, ChIP-seq, and ATAC-seq reported in this paper have been deposited in the Genome Sequence Archive (GSA) for human under accession number HRA000113 and Gene Expression Omnibus (GEO) database under the accession number: GSE146901. GEO accession codes of the published data used in this study are as follows: CTCF ChIP-seq of CD4+ T cell and Jurkat cell line, GSE12889; CTCF ChIP-seq of Loucy cell line, GSE123214; sATAC-seq of CD4+ T cell, GSE87254; ATAC-seq of Jurkat cell line, GSE115438; H3K27ac ChIP-seq of CD4+ T cell, GSE122826; H3K27ac ChIP-seq of Jurkat cell line, GSE68978; H3K27ac ChIP-seq of Loucy cell line, GSE74311; RNA-Seq of Loucy cell line, GSE100694; RNA-seq of T-cell development, GSE69239. Graph files for the called peaks have been deposited on UCSC [https://genome.ucsc.edu/s/ChenFengling/TALL]. The remaining data are available within the article, Supplementary Information or available from the authors upon request. Source data are provided with this paper.

## Code availability
All code for Hi-C analysis have been deposited on GitHub (https://github.com/ChenFengling/HiCpipe).

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

## Acknowledgements

We thank Drs Meng Lv, Yingjun Chang, and Yan Chang for sample collection; Dr Cheng Li of Peking University for critically reviewing the manuscript. We also thank Yan Liu, Fei Wang, and Xuefang Zhang from the National Center for Protein Sciences Beijing at Peking and Tsinghua Universities for assistance with FACS. This project was supported by the Peking-Tsinghua Center for Life Sciences, Beijing Advanced Innovation Center for Genomics at Peking University for HW and the National Natural Science Foundation of China (81602254 for L.Y., 31871343 for Y.C., 31671384 and 81890994 for Y.C. and M.Q. Z.). F.C. and W.W. were supported by the Postdoctoral Fellowship of Peking-Tsinghua Center for Life Sciences.

## Author contributions

L.Y., H.Z., and H.W. conceived the project; Y.C. designed the Hi-C and RNA-seq experiments; L.Y., H.Z., M.S., and W.W. performed the Hi-C and RNA-seq experiments; F.C. designed the bioinformatic pipelines and performed the Hi-C and RNA-seq integrated analyses, while B.D. conducted the survival analysis. Q.J., L.Z., and X.H. contributed the clinical samples and data. L.Y., F.C., and B.D. generated the figures and tables. L.Y., F.C., and H.W. wrote the manuscript with help from all authors. X.H. was in charge of the clinical study; M.Q.Z. and Y.C. oversaw the bioinformatics analyses, and H.W. supervised the entire project.

## Competing interests

The authors declare no competing interests.
