## [Peer Review File · Nature Communications]

REVIEWER COMMENTS

Reviewer #1 (Remarks to the Author): Expert in TAD epigenetics

Musa Mhlanga

Reviewer #3 (Remarks to the Author): Expert in leukemia epigenetics

The manuscript by Yang and colleagues describes the 3-D chromatin architecture within blast cells from 18 newly diagnosed T-ALL patients. The study, whilst interesting, is entirely descriptive and the very recent paper by the Tsirigos lab published in Nature Genetics (March 23, 2020) decreases the novelty of the Yang manuscript. The integration of this data for structural (both 3-D looping and translocations) and gene expression changes is a strength of the manuscript, however, the authors need to avoid the temptation to describe these biological phenomena as cause-effect relationships or propose mechanisms which are not empirically tested. Demonstrating that 3D structure can discriminate between different sub-types of T-ALL is an interesting observation – but it does not provide new insights that define novel clinical classifications or drug targets. Furthermore, describing complex 3D interactions in simple ways is hugely challenging, and while the authors have addressed this to an extent, the complexity of some of the figures and the technical jargon make it difficult in places to get a clear understanding of what the authors are trying to summarise or conclude. Clarity in this respect is important to increase accessibility of the study for the non-expert general readership of Nature Communications. My specific concerns are listed below.

1. Supplemental figure 1b and text line 66. It's unclear what is meant by A-to-B and B-to-A changes. Please clarify.
2. Text lines 64-70. This text lists a series of features which, as written, is cumbersome. Please streamline, the details are less important, than the overall picture.
3. Figures 1a,b. It is not surprising that clustering can define subtypes of T-ALL at the structural or gene expression level. However, the authors need to be careful in interpreting how the two are linked in cause-effect relationships based on the clustering data. Text lines 71-76 need to be re-worked so that you do not lead the reader to think that cause-effect relationships have been established.
4. The GO ontology analyses of Figures 1a and 2d should be moved to supplementary information. They do not add significantly to the story.
5. The relevance of the CTCF and H3K27ac profiles described in figures 1f,g and 2 g,h and elsewhere need to be spelled out clearly. What are their possible functional roles in mediating changes in looping interactions?
6. Please define what is meant by the term “structure alteration-mediated dysregulated transcription factors?”
7. Figure 2a. I do not see how this figure depicts the relationships between T-ALL samples and developmental stages. This needs to be described better or depicted differently in the figures. The developmental stages need to be clearly defined in the legend or text.
8. The concept of enhancer hijacking is a valid hypothesis – but again the authors need to tone down the text as they do not empirically demonstrate that the enhancers are driving the ectopic expression of the genes in question. CRISPR-based approaches where the enhancer is deleted or

mutated could be used to address this question – but this may be beyond the scope of this manuscript.

9. Figure 4 and text lines 180-194. There is a bit of a disconnect with the focus of this part of the study. The authors acknowledge that it is not known whether HOX-positive T-ALLs represent a homogeneous clinical entity. However, it's not clear to this reviewer how discriminating the difference between HOXA- and HOXA+ T-ALLs shed light on this in any way – as this is done at the molecular analysis, and the data is not put into a clinical context. Furthermore, the sample sizes the authors use here, do not lend themselves to a rigorous discrimination between the two different molecular subtypes of HOXA+ T-ALL. More samples would need to be analysed to improve the statistical rigor of the study.

10. Text lines 233-244. Again, the authors assume cause and effect between new loop formations with enhancers and ectopic expression. There is no mechanistic proof that this is the case. This text needs re-wording.

11. The relationship between prognosis and HOXA expression is interesting, but the authors make no attempt to link this to the 3-D structures. Therefore, the data shown in Figure 6 is out of step with the rest of the manuscript. This is a missed opportunity and the manuscript could be strengthened if prognosis and 3-D structures could be associated statistically. The authors should be encouraged to examine this further.

12. In line with my comments above, I think the title of manuscript is misleading as mechanisms have not been proven and 3D structures have not been shown to have prognostic value.

Reviewer #4 (Remarks to the Author): Expert in Hi-C

In this work, the authors performed Hi-C in 18 T-ALL samples and 4 normal controls. Then they performed a series of standard Hi-C data analysis, including A/B compartment, TADs, chromatin loops. All the results seem to be expected: for example, clustering analysis can separate T-ALL from EPT ALL and control. There are certain groups of genes that are up or down-regulated and they are located in the A/B compartment switching regions.

Most important, a key paper was not mentioned - Kloektgen et al, Nature Genetics 2020, "Three-dimensional chromatin landscapes in T cell acute lymphoblastic leukemia". In this work, the experiment design was exactly the same as this manuscript - Hi-C in over 10 T-ALL and EPT ALL patients. The authors need to demonstrate convincingly what novelty is compared with Kloektgen et al. It is also puzzling how the author missed to cite this paper, as the biorxiv version has been out there since last year. Therefore, according to this reviewer, the novelty of this work is not convincing, and several critical experiments are needed.

1. Introduction is poorly written and missed all the seminal work in this field, such as enhancer-hijacking (Northcott et al, Weischenfeldt et al.) and neo-TADs (Mundlos lab).
2. Enhancer: The authors need to perform CHIP-Seq for enhancer marks to support the predicted enhancer-hijacking. It's not clear in the writing, but it seems the enhancers are potentially from cell lines and directly been used as enhancers in patient samples.
3. Minimally, they need to perform reporter assay to validate their "hijacked" enhancers are really enhancers.
4. CNVs were not considered in this paper. They authors need to perform WGS to confirm that the up-regulation/down-regulation of the genes were not due to CNVs.

5. No validation for the predicted enhancer-gene pairs. The authors need to perform either CRISPR/Cas9 or CRISPRi to silence the hijacked enhancer and observe the effect on the target genes.

We appreciate the positive comments and critical suggestions from all three reviewers. We have conducted additional experiments as suggested by the reviewers, which made our conclusions much solid. We also revised the entire manuscript according Nature Communications format. Below are our point-by-point responses to reviewers' insightful critiques (in blue type).

Reviewer #1 (Remarks to the Author): Expert in TAD epigenetics (Musa Mhlanga)

In this manuscript the authors use BL-HiC on 18 patients with T-lineage acute lymphoblastic leukemia (T-ALL) and 4 healthy T cell controls. They find significant alterations in the 3D genome structure of T-ALL patients at the level of TADs, breakpoints (and translocations in cis and trans) which implicate abrogated looping and enhancer hijacking. As a result of these processes, the majority of which occurred in noncoding regions of the genome, Gene expression is significantly altered. The authors focus on the alterations in genome structure in the HOXA cluster identifying overexpression of 3 HOXA cluster genes to be associated with poor outcomes.

The authors make highly interesting findings including A to B or B to A compartment switches in TADs caused by 3D genome alterations. Switches from B to A compartment were mostly accompanied by upregulation of genes while switches from A to B were associated with the silencing of genes. This analysis led to the identification of the potential target of CDK6 as a potential target for T-ALL treatment as it was found to be upregulated in all T-ALL samples regardless of other classifications. Importantly, 3D Genome analysis could distinguish two previously identified sub classes of T-ALL namely ETP and non ETP. The authors speculate that this may indicate two developmentally frozen states of T cell development. The authors then focus on the a further classification of non ETP T-ALL into HOXA, TLX and TAL subtypes. There they focus on the more "trans" mechanism involving translocation mediated fusion which all epigenetically activate the HOXA cluster. A significant part of their findings and analysis focus on this classification. It is in this section that they identify enhancer hijacking to be an especially important mechanism for the ectopic activation of HOXA11-A13.

Overall this study in its use of 3D genomic alterations to classify and study the prognosis of multiple patients with T-ALL is of very high interest to both the basic

scientific community and eventually from clinicians. The manuscript is well written and experiments are in general well controlled. The authors support their claims with good data and remarkably are able to partition T-ALL types by 3D Genome reproducing clinical designations. This manuscript is appropriate for a journal like Nature Communications. Some suggestions for the authors below:

Major points:

In the section between line 104 and 122, the authors studied loop formation between ETP and non-ETP ALL. But, they did not use normal cells at the same T cell developmental stages. Therefore, the loops may be caused due to either developmental stage, ETP ALL (or non-ETP) specifically or sample specifically. In other words, we do not know this difference of the loops is caused by developmental stage or tumorigenesis. I think they should check loop formation in normal cells such as Thy1 stage and Thy3 stage by 3C (or other 'C' technologies such as Hi-C). This would strengthen their claims. If this loop is caused by development, it is strange that some oncogenic genes are upregulated.

We fully agree with the reviewer on this important point. T-ALL is associated with T cell differentiation arrest at different stages of thymocyte development, and therefore, the proper comparison should be drawn from normal T cells at the same developmental stages. However, early T cell development happens in the thymus. It is almost impossible for us to get thymus from healthy donors to check loop formation at Thy1 and Thy3 stages. Probably because of the same reason, there are not published Hi-C data for human T cells at early developmental stages and recent publication on 3D chromatin landscapes of T-ALL (Kloetgen et al 2020 Nature Genetics¹) also used normal T cells isolated from peripheral blood for comparison. We did add a sentence in the results (page 5, lines 111-113) and discussion section (page 17, lines 364-368) to point out this caveat.

Minor points:

1. At line 93, authors wrote 'While the majority of the DEGs were associated with structural changes at 1 or 2 levels, transcription factor SOX4, WT1 and TFDP2 had structural alterations at all 3 levels'. What does level mean in this context.

We apologize for this unclear description. The 3 levels are compartment, TAD and loop chromatin structures, from large to small scale, which can be identified through Hi-C analysis. We found most of the DEGs are associated with 1 or 2 such structural changes. However, transcription factor *SOX4*, *WT1* and *TFDP2* had B to A compartment change, increased D-score and enhanced loops in certain T-ALL samples. We have deleted this paragraph and also added a brief introduction of hierarchical 3D genome organization in the Introduction section (page 3 lines 59-72) to avoid confusion.

2. At line 104, the difference between ETP and non-ETP ALL should be in the introduction section. The authors would do well to include subheadings in their manuscript to delineate sections of their results

This work was originally submitted to Nature Genetics as a letter, and was directly transferred to Nature Communications without revision. We have now added the Introduction section, including the description about ETP and non-ETP ALL (page 3, lines 50-54), as well as subheadings in this revised manuscript according to the reviewer's suggestion.

3. In the section between line 104 and 122, the authors projected their data to T cell development trajectory. Because not all readers are familiar with T cell development, some people may be confused at this point. For example, one may think what Thy1 and Thy2 are, and which is developmentally early stage. I suggest adding a scheme of T cell development in the Supplementary figure or even in a main figure.

We appreciate the reviewer's insightful suggestion and have added a scheme of T cell development in revised Fig. 2a.

4. In the last section, they studied the relationship between clinical output and expression of HOXA genes by using cohort data in companion paper. This is not surprising because several studies have been published that HOXA genes may

be good prognosis markers. It is still acceptable to include this in the paper. Are there any loop interactions that are better prognosis markers than expression? If they do exist that may be very significant for this paper.

Previous studies have described the association of *HOXA* overexpression and poor outcomes without demonstrating the relationship between genes within the *HOXA* cluster and the clinical phenotypes^{2,3}. By analyzing 3D genome alterations associated with the *HOXA* cluster, we found that chromosomal rearrangements can reshape the loop structures of *HOXA* locus in T-ALL by “cis” (enhancer-hijacking) and “trans” (oncogenic fusion events) mechanisms, leading to differential *HOXA* gene expressions. Furthermore, by studying the association between 3D genome alteration associated *HOXA* gene expressions and clinical phenotypes, we found that T-ALL cases with ectopic expression of the *HOXA11-A13* genes, but not *HOXA1-10*, are enriched for ETP immunophenotype, have higher rate of JAK-STAT mutations and poor outcome.

We agree with the reviewer that identifying loop structures that are specifically associated with poor prognosis could increase the significance of our study. In our original submission, we identified neo-loop formation in 3 *HOXA13* translocation and one *NUP98* fusion cases, which were associated with poor prognosis. As shown in revised Figure 4e, we have expanded our analysis to a larger cohort of *HOXA*+ T-ALL samples and found that *NUP98*-related fusion events and *HOXA-13* translocations are both tightly associated with ectopic expression of the *HOXA13* gene. *HOXA13* positiveness, which can be identified either by gene expression or cytogenetic analysis of translocation and fusion events identified in this study, may serve as a biomarker for identification of T-ALL patients with poor prognosis.

Reviewer #3 (Remarks to the Author): Expert in leukemia epigenetics

The manuscript by Yang and colleagues describes the 3-D chromatin architecture within blast cells from 18 newly diagnosed T-ALL patients. The study, whilst interesting, is entirely descriptive and the very recent paper by the Tsigos lab published in Nature Genetics (March 23, 2020) decreases the novelty of the Yang manuscript. The integration of this data for structural (both 3-D looping and

translocations) and gene expression changes is a strength of the manuscript, however, the authors need to avoid the temptation to describe these biological phenomena as cause-effect relationships or propose mechanisms which are not empirically tested. Demonstrating that 3D structure can discriminate between different sub-types of T-ALL is an interesting observation – but it does not provide new insights that define novel clinical classifications or drug targets. Furthermore, describing complex 3D interactions in simple ways is hugely challenging, and while the authors have addressed this to an extent, the complexity of some of the figures and the technical jargon make it difficult in places to get a clear understanding of what the authors are trying to summarize or conclude. Clarity in this respect is important to increase accessibility of the study for the non-expert general readership of Nature Communications. My specific concerns are listed below.

We appreciate this reviewer's critical comments. Although our experimental designs are very similar to Kloetgen et al, and both works demonstrate that global 3D genome architecture can separate normal T cells and two T-ALL subtypes, ETP and non-ETP ALL, the emphasis and the take-home message of our work are quite different from those published in the Nature Genetics paper.

1. While Kloetgen et al¹ focused on NOTCH pathway regulated genomic structures and recurrent TAD fusion events that regulating *MYC* expressions, we focused our attention on the translocation events previous unknown to the field, especially those translocations within the non-coding regions of the genome.
2. The Nature Genetics paper draw conclusions mainly based on 4 non-ETP T-ALL patient samples and 2 cell lines, we analyzed 18 primary patient samples (8 ETP and 10 non-ETP samples) and utilized BL-Hi-C. These not only allowed us to explore 3D genome alterations on the loop level, but also give us more statistic power to analyze the differences between ETP and non-ETP ALLs and among different non-ETP subtypes.

We also agree that without experimental proofs, we ought to be careful in drawing any conclusion regarding cause-effect relationships. In this revised manuscript, we thoroughly checked and rephrased the overstated sentences.

As for the clinical concerns, by studying the association between 3D genome

alterations and clinical phenotypes, we found that T-ALL cases with ectopic expression of the *HOXA11-A13* genes, but not *HOXA1-10*, are enriched for ETP immunophenotype, higher rate of JAK-STAT mutations and poor outcome. Therefore, *HOXA11-A13* positiveness may serve as a biomarker for identification of T-ALL patients with poor prognosis and anti-JAK-STAT inhibitor treatment may benefit this group of patients.

This manuscript was originally submitted to Nature Genetics as a letter then directly transferred to Nature Communications without reformat. Many important background information was missing from our original manuscript. We have now revised entire manuscript with an Introduction section to avoid technical jargons and aid clear understanding of our study by broad spectrum of readers.

1. Supplemental figure 1b and text line 66. It's unclear what is meant by A-to-B and B-to-A changes. Please clarify.

We have clarified this point in the Introduction section (page 3 lines 61-65).

2. Text lines 64-70. This text lists a series of features which, as written, is cumbersome. Please streamline, the details are less important, that the overall picture.

We have rewritten this paragraph (page 5 lines 106-111).

3. Figures 1a,b. It is not surprising that clustering can define subtypes of T-ALL at the structural or gene expression level. However, the authors need to be careful in interpreting how the two are linked in cause-effect relationships based on the clustering data. Text lines 71-76 need to be re-worked so that you do not lead the reader to think that cause-effect relationships have been established.

We appreciate this reviewer's suggestion and have rewritten this paragraph to avoid potential misleading conclusion (page 6 lines 115-131).

4. The GO ontology analyses of Figures 1a and 2d should be moved to supplementary information. They do not add significantly to the story.

We agree that the results from GO ontology analysis are more or less confirming the knowledge in the field and have now moved these results to supplementary Figures 1 and 2.

5. The relevance of the CTCF and H3K27ac profiles described in figures 1f,g and 2g,h and elsewhere need to be spelled out clearly. What are their possible functional roles in mediating changes in looping interactions?

The CTCF binding site is enriched in loop anchors and H3K27ac marks active enhancer. As loops involve interactions between CTCF-binding sites and/or between enhancers and promoters, we mapped CTCF and H3K27ac profiles of normal T cells and T-ALL samples to provide independent support for the differential loop calling in these analyses.

Considering the reviewer's question, we did a genome-wide analysis and found T-ALL-specific loops were bound by T-ALL-specific CTCF or H3K27ac and vice versa (revised Supplementary Fig. 1g). A previous study⁴ has demonstrated the instructive function for CTCF in chromatin looping based on the CTCF degron system, which may explain the correlation between CTCF dynamics and loop changes seen in our study. For the correlation between H3K27ac dynamics and loop changes, we speculate that chromatin-associated epigenetic environment may facilitate the loop formation.

6. Please define what is meant by the term "structure alteration-mediated dysregulated transcription factors?"

"structure alteration-mediated dysregulated transcription factors" means those transcription factors whose dysregulated gene expressions are associated with detectable chromatin structural alteration, such as compartment, TAD or loop

changes. We have deleted this sentence from the revised manuscript to avoid potential cause-effect indication.

7. Figure 2a. I do not see how this figure depicts the relationships between T-ALL samples and developmental stages. This needs to be described better or depicted differently in the figures. The developmental stages need to be clearly defined in the legend or text.

We apologize for this unclear description and have added the scheme of T-lineage differentiation in the revised Fig. 2a, in which each development stage is clearly indicated.

8. The concept of enhancer hijacking is a valid hypothesis – but again the authors need to tone down the text as they do not empirically demonstrate that the enhancers are driving the ectopic expression of the genes in question. CRISP-R based approaches where the enhancer is deleted or mutated could be used to address this question – but this may be beyond the scope of this manuscript.

We agree that enhancer hijacking hypothesis needs to be empirically tested and have tried the CRISP-R based approaches as suggested by the reviewer. However, we have encountered some technical obstacles, which prevented us to achieve the goal:

1. None of the established T-ALL cell lines harbors *HOXA13*-related translocations, which forced us to perform mutagenesis either *in vivo* in xenograph models or *in vitro* in the tissue culture system.
2. We tried to culture primary T-ALL cells from case 077, as samples from cases 076 and 108 were used up for Hi-C analysis. We cultured T-ALL blasts according to 2 published protocols^{5,6}. Unfortunately, T-ALL cells could not be expanded. Similar situation has also been documented in the published work⁵.
3. We also tried to establish *in vivo* xenograph model so that the T-ALL cells can be expanded *in vivo*. Although we have successfully generated more than 20 xenograph models from non-ETP samples, we could not generate similar models from the ETP samples, including case 077. This is likely due to the requirement of thymic microenvironment for the growth and survival of early T progenitors, which is lacking in the NSG mice used for generating xenograph models.

Nevertheless, we have tried other approaches to demonstrate the activities of those predicted enhancers identified by our Hi-C analysis, as reported by previous “enhancer hijacking” studies (Northcott et al, 2014, *Nature*⁷; Weischenfeldt et al, 2016, *Nature Genetics*⁸; Zimmerman et al, 2017, *Cancer Discovery*⁹). First, we performed H3K27ac CHIP-seq on case 077 bone marrow samples and found that the predicted enhancers were indeed marked by the H3K27ac mark, an indication for active enhancer (revised Fig. 5b). We then cloned the DNA fragments corresponding to the possible enhancers predicted by Hi-C neo-loops from all 3 cases and conducted luciferase reporter assays. These putative enhancers led to robust reporter activities as compared to the control fragments (see revised Fig. 5d). Moreover, in our study, the resolution of genome contact maps are accurate to loop level, and we predicted all of the enhancers based on “neo-loop” detected with Hi-C. Thus, we provided the direct evidence of physical interactions between genes and their “hijacked” enhancers. Although these additional evidences are very supportive to our hypothesis, it is not definitive proof. We therefore have tone down our conclusion as suggested by the reviewer (page 14 lines 303-333).

9. Figure 4 and text lines 180-194. There is a bit of a disconnect with the focus of this part of the study. The authors acknowledge that it is not known whether HOX-positive T-ALLs represent a homogeneous clinical entity. However, it’s not clear to this reviewer how discriminating the difference between HOXA- and HOXA+ T-ALLs shed light on this in any way – as this is done at the molecular analysis, and the data is not put into a clinical context. Furthermore, the sample sizes the authors use here, do not lend themselves to a rigorous discrimination between the two different molecular subtypes of HOXA+ T-ALL. More samples would need to analyze to improve the statistical rigor of the study.

We agree with the reviewer that this sentence is a bit of disconnect with the focus of this part of the study and have moved it to Figure 6 where we focus on the associations of *HOXA*⁺ T-ALLs and clinical phenotypes.

We also agree with the reviewer that although our Hi-C analysis identified two molecular subtypes of *HOXA*⁺ T-ALLs by fusion and translocation events, the sample

size, especially 5'*HOXA*⁺ sample size was too small to make a concrete conclusion. We have further verified these results by including a larger cohort of *HOXA*⁺ T-ALL samples. Out of 131 *HOXA* positive T-ALL samples (82 from St. Jude, 39 from in house data and 10 from our study), we found 41 with fusion events detected in this study and 4 with *HOXA13-T*. *MLL*-related fusions were clearly associated with the 3'*HOXA* expression pattern (29/31), while *NUP98*-related fusions (5/6) and *HOXA13-T* (4/4) were clearly associated with *HOXA13* expression, a signature for 5'*HOXA* expression pattern (Fig.4e; Supplementary Fig. 4d).

10. Text lines 233-244. Again, the authors assume cause and effect between new loop formations with enhancers and ectopic expression. There is no mechanistic proof that this is the case. This text needs re-wording.

We have re-worded this paragraph to avoid misleading impression of cause-effect relationships (page 14 lines 303-333).

11. The relationship between prognosis and *HOXA* expression is interesting, but the authors make no attempt to link this to the 3-D structures. Therefore, the data shown in Figure 6 is out of step with the rest of the manuscript. This is a missed opportunity and the manuscript could be strengthened if prognosis and 3-D structures could be associated statistically. The authors should be encouraged to examine this further.

We do agree with the reviewer that we did not link the 3-D structural alterations and prognosis directly in our initial submitted manuscript because 18 samples are too small for statistic significant calling. By expanding our analysis on a larger cohort of *HOXA*⁺ T-ALL samples, we now demonstrate that the 5'*HOXA* gene expression pattern is strongly correlated with enhanced 5'TAD contacts in samples with *NUP98*-related fusions, or “neo-loops” formation of 5'*HOXA* gene in samples with *HOXA13* translocations. The ectopic expression of 5'*HOXA* genes, including *HOXA11-A13*, are tightly associated with ETP immunophenotyped, higher rate of *JAK-STAT* mutations and poor prognosis.

The aim of our 3D chromatin structure analysis is to provide mechanistic insights into dysregulated gene expression. Although Hi-C analysis is costly and technically

challenging, the genomic alterations identified by our Hi-C study, such as *HOXA13* translocation and *NUP98*-related fusions should be easily identifiable by cytogenetic analysis, which may serve as biomarkers for ectopic *HOXA11-A13* expressions and poor prognosis.

12. In line with my comments above, I think the title of manuscript is misleading as mechanisms have not been proven and 3D structures have not been shown to have prognostic value.

We have now revised the title as “3D Genome Alterations Associated with Dysregulated *HOXA13* Expression in High-Risk T-Lineage Acute Lymphoblastic Leukemia”.

Reviewer #4 (Remarks to the Author): Expert in Hi-C

In this work, the authors performed Hi-C in 18 T-ALL samples and 4 normal controls. Then they performed a series of standard Hi-C data analysis, including A/B compartment, TADs, chromatin loops. All the results seem to be expected: for example, clustering analysis can separate T-ALL from EPT ALL and control. There are certain groups of genes are up or down-regulated and they are located in the A/B compartment switching regions.

Most important, a key paper was not mentioned - Kloektgen et al, Nature Genetics 2020, “Three-dimensional chromatin landscapes in T cell acute lymphoblastic leukemia”. In this work, the experiment design was exactly the same as this manuscript - Hi-C in over 10 T-ALL and EPT ALL patients. The authors need to demonstrate convincingly what novelty is compared with Kloektgen et al. It is also puzzling how the author missed to cite this paper, as the biorxiv version has been out there since last year. Therefore, according to this reviewer, the novelty of this work is not convincing, and several critical experiments are needed.

We thank the reviewer for the critical comments. We did notice the work of Kloektgen et al right after it appeared on BioRxiv, but the paper was not published when we submitted our work to Nature Genetics. As papers published on BioRxiv are not peer-reviewed, we were not sure whether it would be appropriate to cite this paper when preparing our submission. Now the work has been published, we have cited and discussed the implication of this paper in our revised manuscript.

Although our experimental designs are very similar to Kloektgen et al, and both works demonstrate that global 3D genome architecture can separate normal T cells and two T-ALL subtypes, ETP and non-ETP ALL, the emphasis and the take-home message of our work are quite different from those published in the Nature Genetics paper.

1. While Kloetgen et al¹ focused on NOTCH pathway regulated genomic structures and recurrent TAD fusion events that regulating *MYC* expressions, we focused our attention on the translocation events previous unknown to the field, especially those translocations within the non-coding regions of the genome.
 2. The Nature Genetics paper draw conclusions mainly based on 4 non-ETP T-ALL patient samples and 2 cell lines, we analyzed 18 primary patient samples (8 ETP and 10 non-ETP samples) and utilized BL-Hi-C. These not only allowed us to explore 3D genome alterations under loop level, but also give us more statistic power to analyze the differences between ETP and non-ETP ALLs and among different non-ETP subtypes.
-
1. Introduction is poorly written and missed all the seminal work in this field, such as enhancer-hijacking (Northcott et al, Weischenfeldt et al.) and neo-TADs (Mundlos lab).

This work was originally submitted to Nature Genetics as a letter and was directly transferred to Nature Communications without reformatting. We have now revised the manuscript according to Nature Communications research article format, including an introduction section.

2. Enhancer: The authors need to perform ChIP-Seq for enhancer marks to support the predicted enhancer-hijacking. It's not clear in the writing, but it seems the enhancers are potentially from cell lines and directly been used as enhancers in patient samples.

We appreciate the reviewer's suggestion. In the initial submission, we used the neo-loops between translocated chromosomes detected on Hi-C heatmaps to predict possible enhancers and used CTCF, ATAC-seq and H3K27ac data from Jurkat cell line to confirm our prediction. Now, we have performed H3K27ac ChIP-seq using blast cells from case 077, as samples from cases 076 and 108 were used up by Hi-C analysis. We have provided evidence that the predicted enhancers are indeed marked by the H3K27ac mark (revised Fig. 5b).

3. Minimally, they need to perform reporter assay to validate their "hijacked" enhancers are really enhancers.

According to the reviewer's suggestion, we perform luciferase reporter assay in the Jurkat cell line, which does not express genes within the *HOXA* cluster. We cloned the DNA fragments corresponding to the putative enhancers predicted by Hi-C neo-loops from all 3 cases and conducted luciferase reporter assays. These putative enhancers led to robust reporter activities as compared to the control fragments (see revised Fig. 5d).

4. CNVs were not considered in this paper. They authors need to perform WGS to confirm that the up-regulation/down-regulation of the genes were not due to CNVs.

We appreciate this reviewer's insightful comment. Since 18 samples size is too small, we have analyzed CNV data of 242 T-ALL samples from Liu et al (Nature Genetics 2017) and identified 110 upregulated genes with copy number gain and 250 downregulated genes with copy number loss, respectively. By alignment of these CNV-related dysregulated genes with the dysregulated genes associated with 3D genome alterations identified in our Hi-C study (568 upregulated and 428 down

regulated), we only identified 1/568 upregulated gene that exhibited copy number gain and 8/428 downregulated genes that exhibited copy number loss. Therefore, the majority of the dysregulated genes identified by the Hi-C study are not associated with CNV changes. We have added this important message in our revised manuscript (page 6 lines 133-140).

5. No validation for the predicted enhancer-gene pairs. The authors need to perform either CRISPR/Cas9 or CRISPRi to silence the hijacked enhancer and observe the effect on the target genes.

We totally agree with this reviewer that enhancer hijacking hypothesis needs to be empirically tested and have tried the CRISPR/Cas9 or CRISPRi to silence the hijacked enhancer as suggested by the reviewer. However, we have encountered some technical obstacles, which prevented us to achieve this goal:

1. None of the established T-ALL cell lines harbors *HOXA13*-related translocations, which forced us to perform mutagenesis either *in vivo* in xenograph models or *in vitro* in the tissue culture system.
2. We tried to culture primary T-ALL cells from case 077 as samples from cases 076 and 108 were used up for Hi-C analysis. We cultured T-ALL blasts according to 2 published protocols^{5,6}. Unfortunately, T-ALL cells could not be expanded. Similar situation has also been documented in the published work⁵.
3. We also tried to establish *in vivo* xenograph model so the T-ALL can be expanded *in vivo*. Although we have successfully generated more than 20 xenograph models from non-ETP samples, we could not generate similar models from the ETP samples, including case 077. This is likely due to the requirement of thymic microenvironment for the growth and survival of early T progenitors, which is lacking in the NSG mice used as recipients for generating xenograph models.

Nevertheless, we have tried other approaches suggested by the reviewer (see our answers to questions 2 and 3) to demonstrate the activities of predicted enhancers identified by our Hi-C analysis, as reported by previous studies (Northcott et al, 2014, *Nature*⁷; Weischenfeldt et al, 2016, *Nature Genetics*⁸; Zimmerman et al, 2017, *Cancer Discovery*⁹). It is worth noting that in our study, the genome contact maps of T-ALL

are accurate to the loop levels, and we predicted all the enhancers based on “neo-loops” between translocated chromosomes, which at least provides the direct evidence for physical interactions between genes and their “hijacked” enhancers. Although these additional evidences are very supportive to our hypothesis, it is not definitive proof. We therefore have tone down our conclusion as suggested by the reviewer (page 14 lines 303-333).

- 1 Kloetgen, A. *et al.* Three-dimensional chromatin landscapes in T cell acute lymphoblastic leukemia. *Nat Genet* **52**, 388-400, doi:10.1038/s41588-020-0602-9 (2020).
- 2 Matlawska-Wasowska, K. *et al.* MLL rearrangements impact outcome in HOXA-deregulated T-lineage acute lymphoblastic leukemia: a Children's Oncology Group Study. *Leukemia* **30**, 1909-1912, doi:10.1038/leu.2016.60 (2016).
- 3 Bond, J. *et al.* An early thymic precursor phenotype predicts outcome exclusively in HOXA-overexpressing adult T-cell acute lymphoblastic leukemia: a Group for Research in Adult Acute Lymphoblastic Leukemia study. *Haematologica* **101**, 732-740, doi:10.3324/haematol.2015.141218 (2016).
- 4 Nora, E. P. *et al.* Targeted Degradation of CTCF Decouples Local Insulation of Chromosome Domains from Genomic Compartmentalization. *Cell* **169**, 930-944 e922, doi:10.1016/j.cell.2017.05.004 (2017).
- 5 Yost, A. J. *et al.* Defined, serum-free conditions for in vitro culture of primary human T-ALL blasts. *Leukemia* **27**, 1437-1440, doi:10.1038/leu.2012.337 (2013).
- 6 Seet, C. S. *et al.* Generation of mature T cells from human hematopoietic stem and progenitor cells in artificial thymic organoids. *Nat Methods* **14**, 521-530, doi:10.1038/nmeth.4237 (2017).
- 7 Northcott, P. A. *et al.* Enhancer hijacking activates GF11 family oncogenes in medulloblastoma. *Nature* **511**, 428-434, doi:10.1038/nature13379 (2014).
- 8 Weischenfeldt, J. *et al.* Pan-cancer analysis of somatic copy-number

alterations implicates IRS4 and IGF2 in enhancer hijacking. *Nat Genet* **49**, 65-74, doi:10.1038/ng.3722 (2017).

- 9 Zimmerman, M. W. *et al.* MYC Drives a Subset of High-Risk Pediatric Neuroblastomas and Is Activated through Mechanisms Including Enhancer Hijacking and Focal Enhancer Amplification. *Cancer Discov* **8**, 320-335, doi:10.1158/2159-8290.CD-17-0993 (2018).

REVIEWER COMMENTS

Reviewer #1 (Remarks to the Author):

The authors have adequately answered all queries.

Reviewer #3 (Remarks to the Author):

Overall, the revised manuscript by Yang and colleagues is greatly improved and has benefitted by substantial revisions of the text that improve clarity and interpretations. The authors have also cited the complementary manuscript by Kloetgen, A. et al. and described their findings in the context of this related manuscript. The authors have addressed all of my concerns where possible.

One minor point: It is difficult to see the names of the TFs in Figure 2e. This figure should be re-drawn to make the annotation more obvious.

Reviewer #5 (Remarks to the Author):

In this revised manuscript the authors have addressed some of the points initially raised by all referee's. One of the major points raised by both Ref #3 and #4 was the overall similitude of this study with that recently published in Nat. Gen. by Kloetgen et al., The authors have answered convincingly that even though the overall design of the study is similar with that of Kloetgen et al, it was conducted on a larger cohort and that the overall focus of the analysis was distinct in their study. Below, here are my comments on the points previously raised by Ref#4. In my opinion the manuscript would be of general interest for the community, should these points and other referee's be raised by the authors.

1. Introduction is poorly written and missed all the seminal work in this field, such as enhancer-hijacking (Northcott et al, Weischenfeldt et al.) and neo-TADs (Mundlos lab).

The introduction was adequately re-written

2. Enhancer: The authors need to perform ChIP-Seq for enhancer marks to support the predicted enhancer-hijacking. It's not clear in the writing, but it seems the enhancers are potentially from cell lines and directly been used as enhancers in patient samples.

The authors have performed ChIP-seq for H3K27ac in 077 sample (ETP, translocation with ERG locus) and could not perform this experiment in other primary samples for understandable reasons. However, and this should clarified in the manuscript, why was the Jurkat ChIP-seq was used as the only reference for H3K27ac signal since 1) it has no expression of the HOXA locus and 2) the observed H3K27ac signal is not always convincing under the putative enhancer detected by Hi-C as for example the cases of E1 and E2 of the 076 case. An option would be to show additionally at least one of the other T cell models (Jurkat, Loucy, 077, KE37 normal T cells) that can show such enhancer signal (see also next point).

3. Minimally, they need to perform reporter assay to validate their "hijacked" enhancers are really

enhancers.

The authors have now performed such assay, presented in Fig. 5b. The justification of the putative enhancer loci chosen for the enhancer assay in Jurkat is not completely clear. These loci were chosen because they establish new loops with the HOXA locus and they presumably harbor H3K27ac signals. However not all loci harbor convincing H3K27ac signal and some other contacts visible in Fig. 5a, b, c might be involved. Were more putative enhancers in the translocated loci tested (and possibly appearing negative)? If so, they should be also displayed.

4. CNVs were not considered in this paper. They authors need to perform WGS to confirm that the up-regulation/down-regulation of the genes were not due to CNVs.

The authors have adequately addressed this question in the revised manuscript.

5. No validation for the predicted enhancer-gene pairs. The authors need to perform either CRISPR/Cas9 or CRISPRi to silence the hijacked enhancer and observe the effect on the target genes.

Although the authors have not addressed this point that would indeed be essential for a demonstration of the hijacking hypothesis, their arguments are receivable.

Additional minor points:

- Since H3K27ac ChIP-seq was performed for 077 sample, it might be useful to include its signal track, for example in Fig. 1f and 2c.
- Panels in Fig. 2e and 2f show HiC map comparison of normal T cell and merged T-ALL. However, visually this comparison is not really fair since the latter have more aligned pairs and thus 3 times more resolution. It should be clearly indicated either in the text, legend or methods so that is completely clear for the reader.
- The supplemental table 2 does not provide the B2A compartment transitions that are described in Fig. 1 (a category called 'other' is present but it is not clear what it covers).

We are pleased to know that reviewers #1 and #3 are satisfied with our revised manuscript and responses. We also tested additional putative enhancer elements and used an additional cell line per reviewer #5's comments. Please find below our responses in blue type.

Reviewer #3 (Remarks to the Author):

One minor point: It is difficult to see the names of the TFs in Figure 2e. This figure should be re-drawn to make the annotation more obvious.

We have re-drawn the Figure 2e and increased the font size and colors to make the annotations more visible.

Reviewer #5 (Remarks to the Author):

In this revised manuscript the authors have addressed some of the points initially raised by all referee's. One of the major points raised by both Ref #3 and #4 was the overall similitude of this study with that recently published in Nat. Gen. by Kloetgen et al., The authors have answered convincingly that even though the overall design of the study is similar with that of Kloetgen et al, it was conducted on a larger cohort and that the overall focus of the analysis was distinct in their study. Below, here are my comments on the points previously raised by Ref#4. In my opinion the manuscript would be of general interest for the community, should these points and other referee's be raised by the authors.

We appreciate the positive comments by this reviewer. In this revised manuscript, we have conducted additional experiments to further clarified the additional points raised by this reviewer.

1. Introduction is poorly written and missed all the seminal work in this field, such as enhancer-hijacking (Northcott et al, Weischenfeldt et al.) and neo-TADs (Mundlos lab).

The introduction was adequately re-written

2. Enhancer: The authors need to perform ChIP-Seq for enhancer marks to support the predicted enhancer-hijacking. It's not clear in the writing, but it seems the enhancers are potentially from cell lines and directly been used as enhancers in patient samples.

The authors have performed ChIP-seq for H3K27ac in 077 sample (ETP, translocation with ERG locus) and could not perform this experiment in other primary samples for understandable reasons. However, and this should clarified in the manuscript, why was the Jurkat ChIP-seq was used as the only reference for H3K27ac signal since 1) it has no expression of the HOXA locus and 2) the observed H3K27ac signal is not

always convincing under the putative enhancer detected by Hi-C as for example the cases of E1 and E2 of the 076 case. An option would be to show additionally at least one of the other T cell models (Jurkat, Loucy, 077, KE37 normal T cells) that can show such enhancer signal (see also next point).

3. Minimally, they need to perform reporter assay to validate their “hijacked” enhancers are really enhancers.

The authors have now performed such assay, presented in Fig. 5b. The justification of the putative enhancer loci chosen for the enhancer assay in Jurkat is not completely clear. These loci were chosen because they establish new loops with the *HOXA* locus and they presumably harbor H3K27ac signals. However not all loci harbor convincing H3K27ac signal and some other contacts visible in Fig. 5a, b, c might be involved. Were more putative enhancers in the translocated loci tested (and possibly appearing negative)? If so, they should be also displayed.

Answers to points 2 and 3:

- a. We appreciate the reviewer's comment and have now added H3K27ac ChIP-seq tracks of 077 and Loucy in revised Fig.5 a-c.
- b. We chose the putative enhancers based on i) new loop formations with *HOXA13* locus as detected by Hi-C (blue circles in revised Fig.5 a-c); ii) harboring H3K27ac signal in sample 077 or T-ALL line Loucy or Jurkat. We actually relied more on H3K27ac signals detected on sample 077 since 076, 108 and 077 are all primary ETP T-ALL samples. As shown in the new enlarged Supplementary Fig.5 c-d with *HOXA13* V4C and trans-loop anchor locations marked, all putative enhancer regions, including E1 and E2 in the case 076, are indeed harbor H3K27ac signal in sample 077.
- c. Although Jurkat does not express the *HOXA* locus, we do find that the putative hijacked enhancer regions are modified by active enhancer marker H3K27ac in Jurkat, as well as primary sample 077 and other T-ALL cell lines, such as Loucy (please see revised Supplementary Fig.5 a-b), suggesting that these putative hijacked enhancers are generally active in T-ALL.
- d. Per reviewer's request, we have tested 6 additional putative enhancers that harbor H3K27ac signals in the trans-loop anchor regions. These results are added in Fig.5d and Supplementary Fig.5c-f.
- e. We appreciate the reviewer's suggestion and have repeated the same enhancer assay on Loucy T-ALL line (see revised Supplementary Figure 5f). Although most of the putative enhancers showed robust reporter activities in both Jurkat and Loucy lines, there are noticeable differences: such as lacking enhancer activities of the E2 and E3 regions in sample 077 in Loucy cells. Although at this moment we do not know the cause of such differences, we do know that Loucy is *HOXA+*, and *HOXA* gene expressions in Loucy cells are trans-activated by fusion protein SET-NUP214 rather than enhancer hijacking.

4. CNVs were not considered in this paper. They authors need to perform WGS to confirm that the up-regulation/down-regulation of the genes were not due to CNVs.

The authors have adequately addressed this question in the revised manuscript.

5. No validation for the predicted enhancer-gene pairs. The authors need to perform either CRISPR/Cas9 or CRISPRi to silence the hijacked enhancer and observe the effect on the target genes.

Although the authors have not addressed this point that would indeed be essential for a demonstration of the hijacking hypothesis, their arguments are receivable.

Additional minor points:

- Since H3K27ac ChIP-seq was performed for 077 sample, it might be useful to include its signal track, for example in Fig. 1f and 2c.

We have now included H3K27ac ChIP-seq track of 077 in Fig.1e and f, Fig. 2f and g, Fig. 3e, Fig.5 a and c, Supplementary Fig. 3c and d and Supplementary Fig. 5.

- Panels in Fig. 2e and 2f show HiC map comparison of normal T cell and merged T-ALL. However, visually this comparison is not really fair since the latter have more aligned pairs and thus 3 times more resolution. It should be clearly indicated either in the text, legend or methods so that is completely clear for the reader.

As we mentioned in Method part that “Hi-C maps of each condition were normalized by its cis interaction pairs”. To avoid misunderstanding, we clarified this point in the legend of Fig.1e of revised manuscript.

To further demonstrate that Hi-C maps of individual samples and merged sample are consistent, Hi-C maps of each individual sample in Fig.1e and 1f are illustrated below.

Hi-C interaction heat maps of *CDK6*

Hi-C interaction heat maps of *SOX4*

- The supplemental table 2 does not provide the B2A compartment transitions that are described in Fig. 1 (a category called 'other' is present but it is not clear what it covers).

We checked the supplemental table 2 and confirmed all compartment switch types were listed, including B2A. According to the reviewer's suggestion, we add the annotation of "Other" in the legend of Fig. 1d. As mentioned in the methods, A or B compartments were defined in each condition by the over 70% sample majority rule. Thus "Other" refers to those compartment states can't be defined in either condition.

REVIEWERS' COMMENTS

Reviewer #5 (Remarks to the Author):

The authors have adequately answered to all the points raised. In my opinion the manuscript can now be published in Nat. Comm.